# Single cell transcriptomics of human epidermis identifies basal stem cell transition states

Shuxiong Wang [1,2,3,8], Michael L. Drummond[1,8], Christian F. Guerrero-Juarez [1,2,3], Eric Tarapore [1], Adam L. MacLean [2,3], Adam R. Stabell[1], Stephanie C. Wu[1], Guadalupe Gutierrez[1], Bao T. That[1], Claudia A. Benavente [1,4,5], Qing Nie[1,2,3,5,6✉] & Scott X. Atwood [1,3,5,6,7✉]

How stem cells give rise to epidermis is unclear despite the crucial role the epidermis plays in barrier and appendage formation. Here we use single cell-RNA sequencing to interrogate basal stem cell heterogeneity of human interfollicular epidermis and find four spatially distinct stem cell populations at the top and bottom of rete ridges and transitional positions between the basal and suprabasal epidermal layers. Cell-cell communication modeling suggests that basal cell populations serve as crucial signaling hubs to maintain epidermal communication. Combining pseudotime, RNA velocity, and cellular entropy analyses point to a hierarchical differentiation lineage supporting multi-stem cell interfollicular epidermal homeostasis models and suggest that transitional basal stem cells are stable states essential for proper stratification. Finally, alterations in differentially expressed transitional basal stem cell genes result in severe thinning of human skin equivalents, validating their essential role in epidermal homeostasis and reinforcing the critical nature of basal stem cell heterogeneity.

[1] Department of Developmental and Cell Biology, University of California, Irvine, Irvine, CA 92697, USA. [2] Department of Mathematics, University of California, Irvine, Irvine, CA 92697, USA. [3] NSF-Simons Center for Multiscale Cell Fate Research, University of California, Irvine, Irvine, CA 92697, USA. [4] Department of Pharmaceutical Sciences, University of California, Irvine, Irvine, CA 92697, USA. [5] Chao Family Comprehensive Cancer Center, University of California, Irvine, Irvine, CA 92697, USA. [6] Center for Complex Biological Systems, University of California, Irvine, Irvine, CA 92697, USA. [7] Department of Dermatology, University of California, Irvine, Irvine, CA 92697, USA. [8]These authors contributed equally: Shuxiong Wang, Michael L. Drummond. ✉email: qnie@uci.edu; satwood@uci.edu

Defining stem cell (SC) heterogeneity and its functional consequences in tissue homeostasis remains an open question in biology. In the skin, SCs reside in the basal compartment of the interfollicular epidermis (IFE) and in discrete compartments within ectodermal skin appendages–namely the pilosebaceous unit and sweat gland[1]. Decades of work primarily from mouse studies have demonstrated the presence of multiple SC pools residing in various compartments of the pilosebaceous unit, including the bulge, hair germ, isthmus, junctional zone, upper portion of the infundibulum, and sebaceous gland[2]. In contrast to the pilosebaceous unit, basal SCs in the IFE are considered more homogenous and are thought to have one or two distinct subpopulations depending on the body site in mouse or human. However, a large degree of plasticity exists within the skin SC populations. Bulge or IFE SCs can both form the pilosebaceous unit and sweat glands in response to inductive signals from the underlying dermis[3], suggesting that IFE SCs may be more heterogenous than previously thought. Whether epidermal SCs exist on a continuum and can equally respond to inductive signals or whether they occupy more stable states is unclear.

IFE self-renewal is thought to be achieved under homeostatic conditions by proliferation of SCs in the basal compartment, followed by transit amplification and terminal differentiation of the SC progeny[2]. Early pedigree studies in mouse dorsal skin suggest that a single basal SC gives rise to transit amplifying (TA) cells with limited proliferation capacity that are destined to undergo terminal differentiation, coined the epidermal proliferative unit (EPU)[4,5]. Other models suggest a single population of committed progenitor cells that directly self-renew or differentiate[6,7], a slower cycling SC population that gives rise to committed progenitor cells that directly differentiate[8], or two independent SC populations that regenerate at different rates[9].

How human IFE self-renewal is achieved is unclear, primarily because of its complex architecture and absence of genetic and imaging tools. Using long-term fate mapping strategies enabled by a lentivirus-mediated gene transfer approach[10], basal SCs appear to be dispersed along the basal compartment in human foreskin epidermis and that EPUs are present and capable of engaging in epidermal self-renewal in human xenografts[11]. This gene transfer approach also enabled the observation that basal SCs do not preferentially occupy specific regions along the basal layer, but rather occupy locations throughout the basal compartment of human skin. However, whether each EPU arose from similar or distinct SC populations remains unclear as the widths and columns of the EPUs varied considerably depending on the originating site.

The advent of single cell RNA-sequencing (scRNA-seq) technologies has enabled the study of cellular heterogeneity, reconstruction of lineage hierarchies, inference of signaling networks, and has partially enabled the dissemination of functional SC roles in complex tissues. scRNA-seq has been successfully applied to normal human tissues including skin epidermis[12] and dermis[13,14], and skin-related pathologies such as nasal polyps[15]. In mice, scRNA-seq has identified extensive functional heterogeneity in skin[16–19], hair follicles[20,21], and regenerative and non-regenerative wounds[22,23]. Despite these studies, epidermal SC heterogeneity of human IFE remains unresolved. To address this issue, we interrogate epidermal cell heterogeneity within human neonatal foreskin epidermis using droplet-enabled scRNA-seq and identify four spatially distinct basal SC subpopulations. Interrogation of the transitional basal subpopulations that spatially occupy both the basal and suprabasal layers indicate their essential role in epidermal homeostasis. Our findings argue against a single population of progenitor cells and suggest a more complex model of multiple epidermal SC transitions that maintain epidermal homeostasis.

## Results

### scRNA-seq identifies heterogeneity in human epidermis.
To define the cellular heterogeneity of human IFE, we isolated viable, single cells from discarded and deidentified human neonatal foreskin epidermis and subjected them to droplet-enabled scRNA-seq to resolve their individual transcriptomes (Fig. 1a; Supplementary Fig. 1; Supplementary Data 1; $n = 5$). We chose foreskin epidermis because it is composed of mostly IFE and contains few rudimentary skin appendages, such as hair follicles and sweat glands[24]. We processed a total of 17,553 cells and performed quality control analysis on individual libraries using Seurat (Supplementary Fig. 2)[25]. We used Similarity matrix-based OPtimization for Single Cell (SoptSC) to bioinformatically parse and analyze our data[26]. We chose the SoptSC algorithm because it is based on a cell-cell similarity matrix that coherently performs many inference tasks under the same framework–including unsupervised clustering, pseudotemporal ordering, cell lineage inference, cell-cell communication, and network inference. SoptSC clustered cells from all five libraries into seven distinct cell communities in an unsupervised manner, corresponding to four distinct cellular cohorts, using Graph embedding (Fig. 1b). A cellular cohort is defined as a group of cell communities expressing similar known marker genes. No significant batch effects were observed upon integration (Fig. 1c). Basal SC communities BAS-I – BAS-IV represented ~4%, ~9%, ~7% and ~3% of the entire population pool, respectively, and were enriched for known basal keratinocyte marker genes including KRT14, KRT5, and CDH3 (Fig. 1d, e). Although known basal marker genes were able to distinguish between cellular identities (i.e., basal vs. granular keratinocytes), they were not sufficient to distinguish between basal clusters despite being clustered distinctly by SoptSC. The spinous community SPN, representing ~54% of the entire population pool, showed heightened expression of KRT1, KRT10, DSG1, and CDH1 that continued to be expressed in granular keratinocytes (Fig. 1d, e). Similarly, spinous marker genes alone were not sufficient to distinguish between spinous clusters, even when they were clustered distinctly by SoptSC. Differentiated granular keratinocytes (GRN, ~16% of total cells) expressed the differentiation gene markers DSC1, KRT2, IVL, and TGM3. SoptSC clustered melanocytes into one single cluster (MEL, ~6% of total cells). This cluster was enriched for the melanocytic markers MITF and MLANA. These results are congruent with clustering of individual libraries (Supplementary Figs. 3 and 4; Supplementary Data 1).

To further corroborate the robustness of SoptSC, we compared its clustering performance with Seurat. To do this, we used the supervised clustering method feature in Seurat, which identified similar cell communities as SoptSC (Supplementary Figs. 3 and 4). For example, library 3 basal SC communities were grouped into three distinct communities for both SoptSC and Seurat (Supplementary Fig. 3A, E). Seurat clustered Langerhans cells separately based on expression of CD207 and CD86 (LAN ~1%) and identified a community composed of erythrocytes (ER ~1%) based on expression of HBB, which were not distinctly resolved using SoptSC (Supplementary Fig. 3a–c, e–g). Two other libraries contained very small clusters of LAN cells but no other library showed a cluster of ER, reinforcing the scarcity of these cells in our dissociation conditions (Supplementary Fig. 4). Dimensionality reduction strategies employed by Seurat and SoptSC remain generally congruent on clustering performance, cell type, and distribution of cell communities (i.e., number of overlapped markers between clusters) across all our individually sequenced libraries (Supplementary Figs. 3e and 4a–h).

### Spatial characterization of keratinocytes in human epidermis.
To define genes associated with basal keratinocytes and human

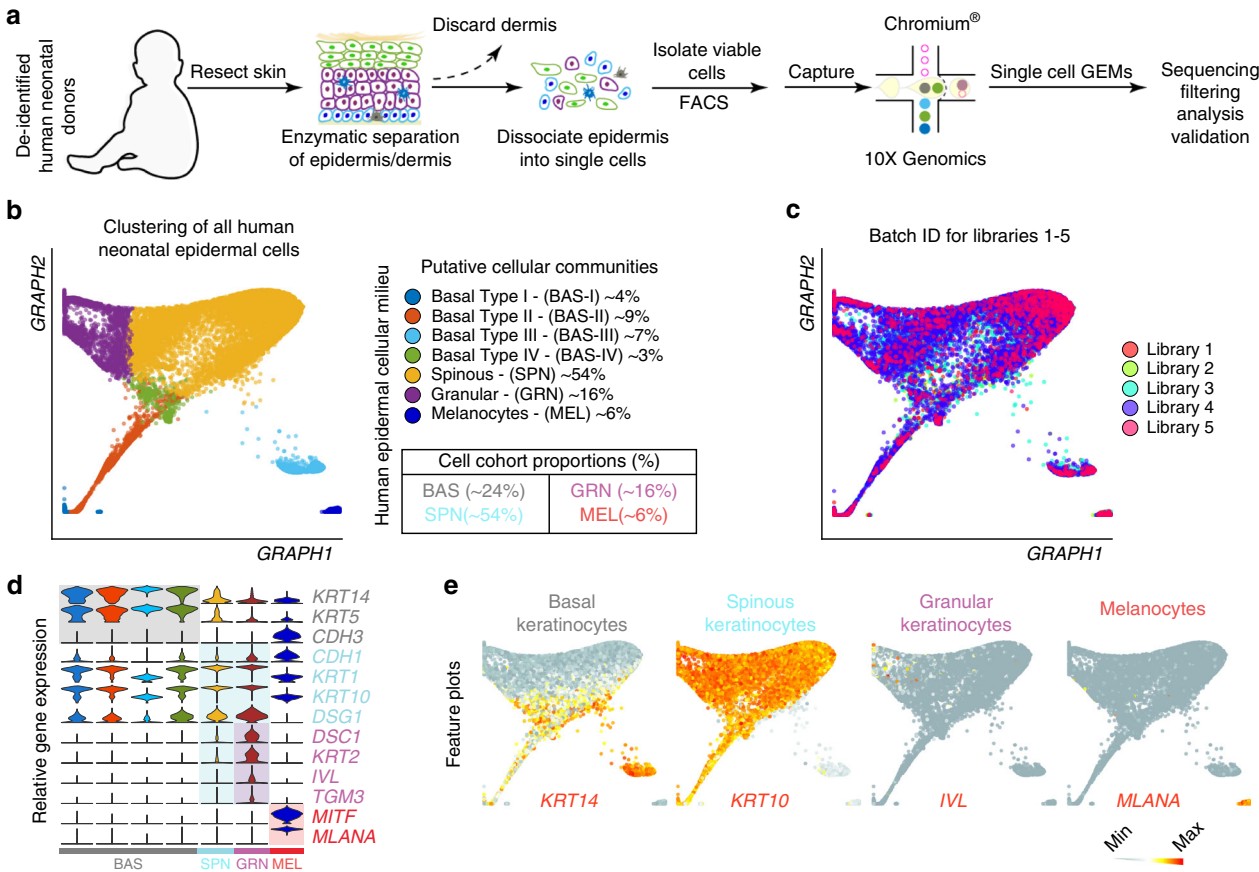

**Fig. 1 Defining human neonatal epidermal cell populations using scRNA-seq. a** Schematic of epidermal cell isolation from human neonatal foreskin. **b** Clustering of 16,360 single cells isolated from Libraries 1-5 that passed quality control metrics using SoptSC and displayed using Graph embedding. Cell proportions from putative cellular communities are quantified on the right. **c** Batch IDs for each library are superimposed onto the Graph embedded clusters and show no substantial batch effects. **d** Violin plots of relative gene expression of known epidermal marker genes split by cell cohorts using SoptSC. **e** Feature plots showing expression of keratinocyte markers from basal, spinous, and granular layers, including melanocytes.

epidermis as a whole, we analyzed differentially expressed genes (DEGs) that define each cluster and found marker genes that provide more specific resolution for each cluster than widely used epidermal marker genes (Fig. 2a–d; Supplementary Data 1). Some marker genes show slight discrepancies depending on the read depth per cell. For instance, the integrated dataset has less reads per cell because of the normalization during batch correction, causing the gap junction gene *GJB2* and *RHCG* to cluster into GRN, whereas they are both clustered into BAS-IV when using library 3 (Fig. 2b, c; Supplementary Data 1). KRT14 immunofluorescence uniformly spans several layers in human neonatal epidermis with KRT10 staining beginning in the second layer and enriching in subsequent layers (Fig. 2e). Basal cluster BAS-III is defined by expression of *ASS1*, *COL17A1*, and *POSTN*, where ASS1 and COL17A1 immunofluorescence staining of neonatal human epidermis shows enrichment between rete ridges, suggesting a specific zone of basal SCs surrounding the papillary dermis (Fig. 2f, l; Supplementary Fig. 5). The BAS-IV basal cluster is defined by expression of *GJB2*, *KRT6A*, and *KRT16*. Immunofluorescence staining of GJB2 shows enrichment at the bottom of the rete ridges with some expression in the upper strata, whereas BAS-III cluster gene KRT19 shows enrichment at the bottom/side of the rete ridges (Fig. 2g, l; Supplementary Fig. 5), reinforcing a specialized zone of basal SCs that can be regenerated after partial-thickness wounding[27]. The BAS-I and BAS-II basal clusters are enriched for cell cycle marker genes but are maintained even after cell cycle regression (Supplementary Fig. 6). The topology of the keratinocyte

subclusters was maintained and cell community gene expression profiles remained congruent with one another, suggesting that cell cycle genes do not profoundly influence keratinocyte subclusters. BAS-I is defined by expression of *PTTG1* and *CDC20*, whereas BAS-II is defined by *RRM2*, *HELLS*, *UHRF1*, and *PCLAF* expression. Immunofluorescence staining of PTTG1, CDC20, PCLAF, and RRM2 show a transitional position within the epidermis where the cells occupy space between the basal and suprabasal layers (Fig. 2h, i, m). Many of these cells are still adjacent to the basement membrane with the bulk of the cell body and nucleus residing either in the basal or suprabasal layers. These "transitional" basal cells appear to be in the process of delaminating from the basal layer, are spread heterogeneously across the epidermis, and may represent basal SCs with a fluid cell fate.

We are also able to identify specific granular keratinocyte genes such as *ZNF750*, *SPINK5*, and *CALML5*, with the latter two showing robust protein enrichment in the granular layer of human neonatal epidermis (Fig. 2j). Melanocyte (*MLANA*) and Langerhans (*CD74*) gene expression signatures also present, with protein expression highly restricted to these cell types (Fig. 2k). However, we are not able to identify specific gene expression signatures that are strongly enriched only in the spinous keratinocytes or that spatially immunostain the spinous layer. The SPN cluster appears to segregate based on lack of basal- or granular-specific markers, suggesting they are at the beginning of a differentiation trajectory that ends with the granular fate and may not be a stable state by themselves.

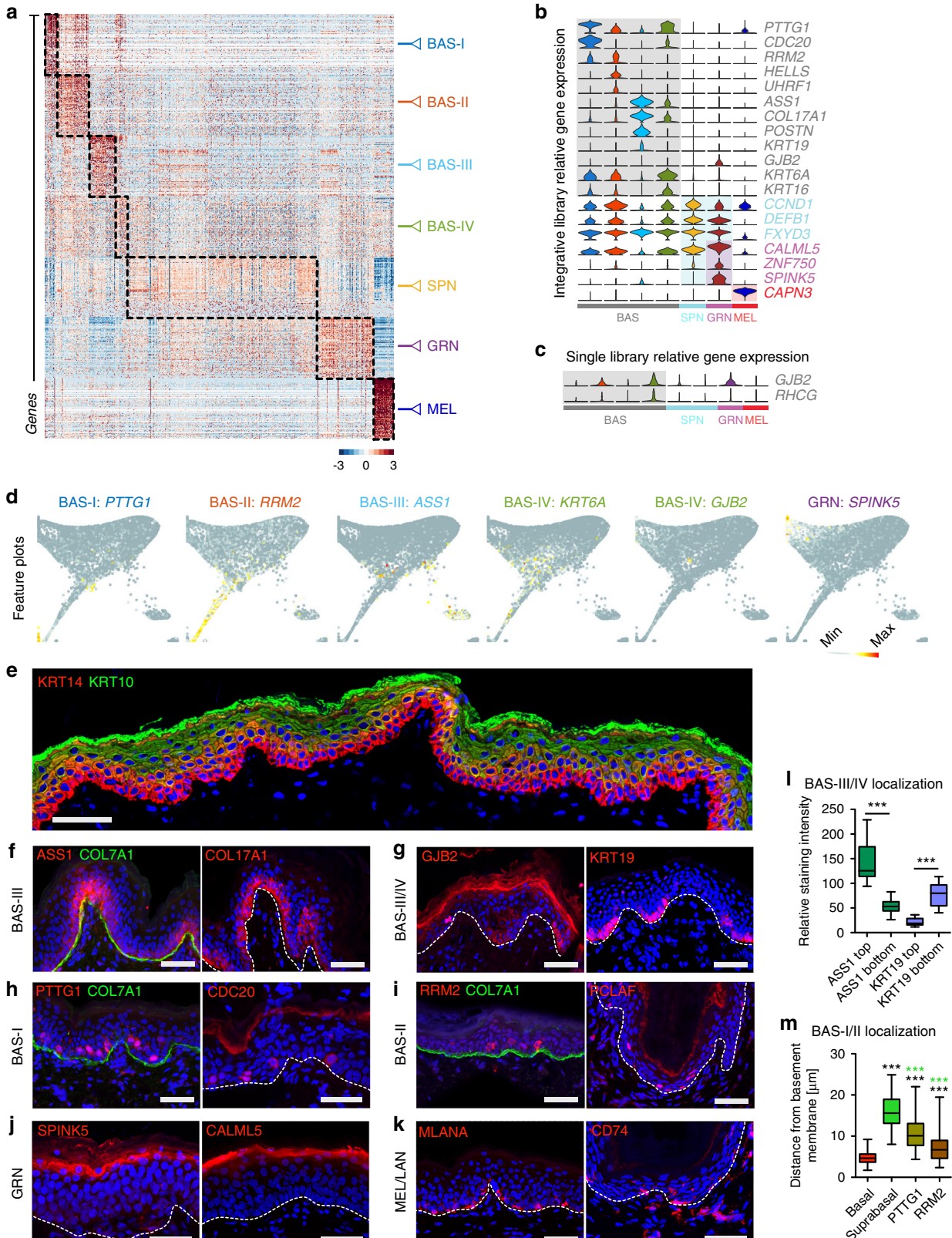

**Cell-cell inference shows cross-talk between keratinocytes**. A feature of single cell analysis is the ability to infer signaling networks within a cell and reconstruct potential cell-cell signaling interactions. Having defined major cell communities in human neonatal epidermis, we sought to quantify potential cell-cell interactions using the probabilistic cell-cell network inference algorithm featured in SoptSC. Signaling relationships in SoptSC are calculated based on differential gene expression and co-variance of specific signaling pathway components. Signaling probabilities between cells are then defined and quantified based on the weighted expression of signaling pathway components between sender-receiver cell pairs inferred via expression of

**Fig. 2 Differential gene expression highlight basal stem cell heterogeneity. a** Heatmap showing top 300 differentially expressed genes per cluster. Dotted lines outline differentially expressed genes. **b** Violin plots of relative expression of marker genes split by cell cohorts and color-coded by cell community as in A. **c** Library 3 violin plots of relative expression of *GJB2* and *RHCG*. **d** Feature plots showing expression of select basal and granular cell marker genes. **e** Immunostaining of KRT14 (red), KRT10 (green), and DAPI (blue) in human neonatal skin. Scale bar 100 μm. $n = 3$. Immunostaining of differentially expressed proteins (red) from the (**f**) BAS-III cluster (ASS1 [$n = 6$] and COL17A1 [$n = 4$]), (**g**) BAS-III/IV clusters (KRT19 [$n = 3$] and GJB2 [$n = 4$]), (**h**) BAS-I cluster (PTTG1 [$n = 6$] and CDC20 [$n = 6$]), (**i**) BAS-II cluster (RRM2 [$n = 6$] and PCLAF [$n = 2$]), (**j**) GRN cluster (SPINK5 [$n = 3$] and CALML5 [$n = 2$]), and (**k**) the MEL (MLANA [$n = 2$]) and LAN (CD74 [$n = 2$]) clusters in human neonatal skin. COL7A1 (green) and DAPI (blue) are costained. Dotted lines denote position of basement membrane where COL7A1 staining is absent. Scale bar 100 μm. **l** Quantification of ASS1 ($n = 14$ ridges) and KRT19 ($n = 10$ ridges) staining intensity at the top and bottom of Rete ridges. ***$p < 0.0001$. **m** Quantification of PTTG1 ($n = 102$ cells) and RRM2 ($n = 104$ cells) distance from the basement membrane, with KRT14 + basal cell ($n = 167$ cells) or KRT10+ suprabasal cell ($n = 142$ cells) distance shown as controls. Black stars represent significance compared with basal cell position whereas green stars represent significance compared with suprabasal position. ***$p < 0.0001$. Box represents 25th to 75th percentiles. Whiskers represent minimum and maximum data points. Bar represents mean. Significance was determined by unpaired two-tailed $t$ test.

---

ligand-receptor pairs and their downstream targets identified from NetPath[28]. We used a reference of known, literature-supported interactions from the WNT, JAK/STAT, NOTCH, and TGF-β signaling pathways and scored interactions between single cells (Supplementary Fig. 7 and Supplementary Data 2)[29,30]. We used Library 3 to generate cell-cell interaction scores because of the high cell count and the greater median gene number per cell (3104 median genes per cell), allowing more ligand-receptor pairs to be quantified than when all libraries are integrated because of normalization from batch correction and showing high interaction score consistency between ligand-receptor pairs among all libraries (Supplementary Fig. 7).

A host of secreted WNT ligands differentially bind to ten distinct Frizzled receptors during canonical WNT signaling to activate Disheveled, subsequently leading to β-catenin stabilization and translocation into the nucleus to turn on WNT-target genes[31]. WNT signaling is a pleotropic signaling pathway heavily involved from the earliest stages of skin development where it helps specify the ectoderm down the skin epithelium lineage[32] and skin appendages[33]. WNT signaling is also involved in adult skin homeostasis where overexpression or loss of WNT/beta-catenin results in a variety of intra- and IFE phenotypes[34,35]. To parse through the cell-cell communication of WNT signaling, we calculated the signaling probability of each ligand-receptor pair and their downstream targets between each cell, averaged their probabilities between each cluster, and hierarchically clustered the aggregate scores to determine the similarity between the cluster-to-cluster interactions associated with each specific pair (Fig. 3a, b). The total cluster level interactions suggest that WNT signaling appears to be active in most of the basal and spinous populations (Fig. 3c). Separating ligand-receptor pairs based on similarity in signaling probabilities identifies more specific signaling networks (Fig. 3d). For instance, WNT4 is restricted to Cluster 7 where the majority of the signaling is directed at BAS-I and SPN-II, recapitulating its basal and suprabasal locations in murine epidermis[35]. Basal SCs in IFE of glabrous skin form an autocrine mechanism important for SC self-renewal, requiring WNT/β-catenin signaling to proliferate and, at the same time, producing and secreting long-range WNT inhibitors to promote differentiation[35]. Deletion of β-catenin in the IFE of adult mice leads to a significant decrease in proliferation, suggesting that the WNT/β-catenin signaling contributes to progenitor cell proliferation under homeostatic conditions[34].

The JAK/STAT pathway is comprised of four nonreceptor tyrosine kinases that are typically activated by cytokine receptors and seven intracellular signaling substrates[36]. Abnormalities in this pathway can cause a number of skin-associated inflammatory disorders such as psoriasis, lupus erythematosus, atopic dermatitis, and alopecia areata. Our predicted JAK/STAT signaling interactions at the cluster level in epidermal keratinocytes show overwhelming activation in granular cells for the Cluster 3 and Cluster 4 signaling networks (Supplementary Fig. 8). Although most of the aforementioned skin disorders are attributable to disruption of JAK/STAT signaling in immune cells, adult epidermis shows strong immunostaining for specific pathway components (JAK3, TyK2, and STAT2/3/4/6) in the *stratum granulosum* layer[37]. In addition, resident skin cells can produce cytokines that help promote skin barrier through cornification (IL-31), lipid envelope composition (IFN-g), and cell-cell adhesion (IL-1a)[38], all processes that predominantly occur in the granular layer.

NOTCH signaling comprises heterodimeric transmembrane receptors where cells expressing any of five NOTCH ligands bind and activate up to four adjacent NOTCH receptors on neighboring cells, thereby initiating cleavage of the intracellular domain and subsequent translocation into the nucleus to help facilitate target gene expression[39]. Psoriasis and all three major skin cancers are associated with disruption of NOTCH signaling. We observe robust activation of NOTCH4 signaling predominantly in granular cells for the Cluster 5 signaling network (Supplementary Fig. 9), suggesting that the granular population may be important receivers of NOTCH signaling and recapitulating known roles of NOTCH in cell fate specification, proliferation, and differentiation[40,41].

The TGF-β family consists of ligands from TGF-β, BMP, Activin, and GDF signaling pathways. These ligands bind to their respective kinase receptors to phosphorylate and activate downstream SMAD effectors to allow their translocation into the nucleus and help facilitate target gene expression[42]. TGF-β also serves as a tumor suppressor, where disrupted TGF-β fuels tumor heterogeneity and drug resistance in squamous cell carcinoma[43]. We observe robust signaling out of the BAS-IV cells and into the BAS-III cells for the Cluster 1, 5, and 6 signaling networks (Supplementary Fig. 10), recapitulating known roles in suppressing basal cell proliferation through the TGF-β and activin ligands[44,45] and suggesting that these populations are uniquely responsive to the TGF-β family. In sum, our cell-cell network inference suggests that epidermal cell communities in human neonatal epidermis can communicate within and between clusters and can recapitulate many of the reported skin-dependent roles of these major signaling pathways. It should be noted that our cell-cell network inference generates hypotheses based upon predictive modeling and does not experimentally validate these events.

**Cell state transitions and pseudotemporal directionality.** Basal keratinocytes undergo terminal differentiation into granular keratinocytes expressing the structural protein Involucrin. During differentiation, keratinocytes progressively lose expression of basal markers, while they concomitantly begin to express

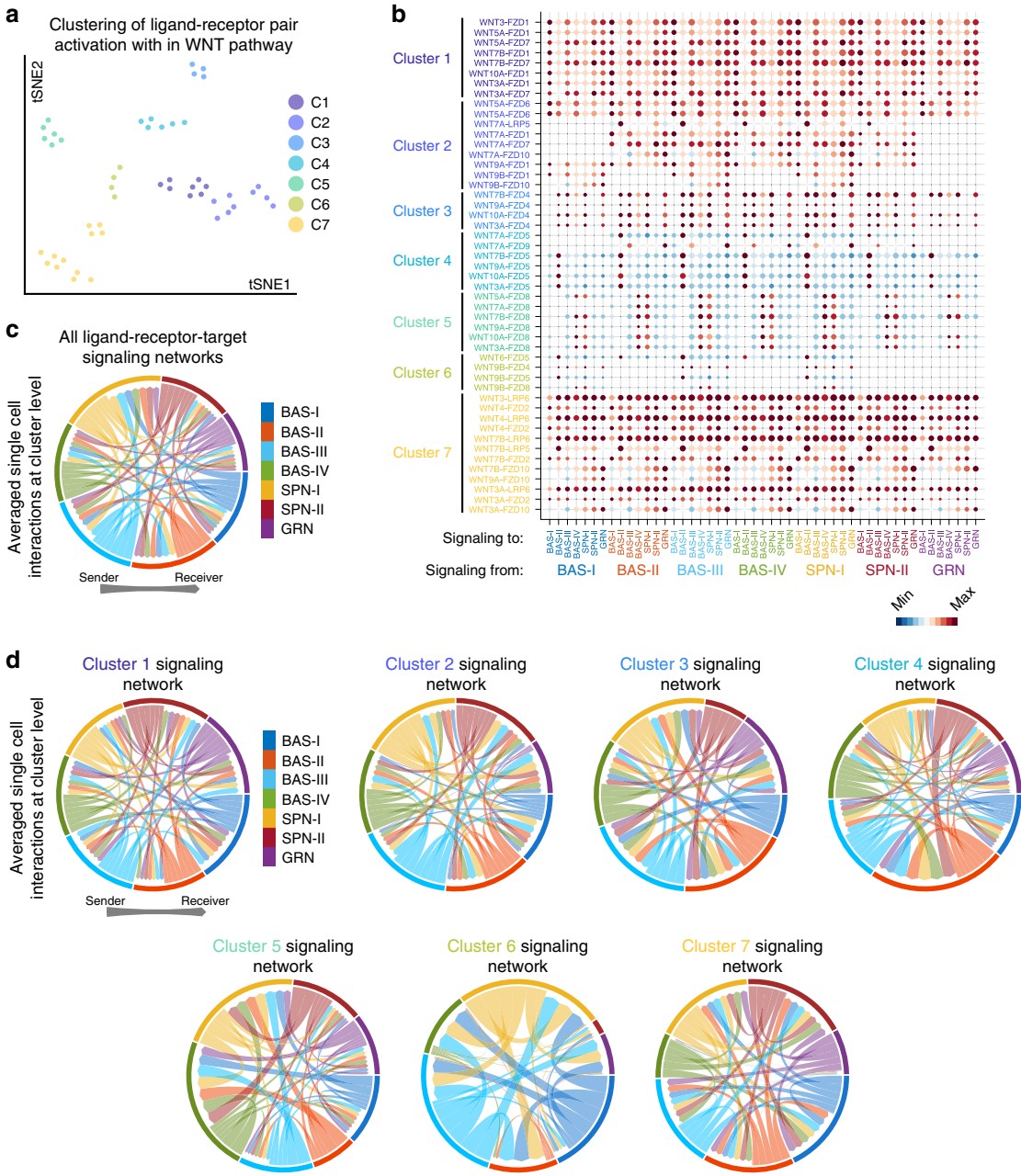

**Fig. 3 Cell-cell interaction modeling of the WNT signaling pathway. a** Hierarchical clustering of similar cell-cell signaling probability scores from library 3 and visualized on a tSNE plot. **b** Visualization of signaling probability scores of Ligand-Receptor pairs and their downstream signaling components. Dot size represents number of averaged cells with probability scores between clusters. **c** Cell-cell communication networks predicted for the entire WNT pathway or for (**d**) individual Cluster networks from B. Edge weights represent the probability of signaling between cell clusters.

suprasal markers. This biological observation enabled us to generate a pseudotime trajectory inferred by SoptSC. We regressed out melanocytes and used the remaining epithelial cohorts composed of BAS, SPN, and GRN communities to model a pseudotemporal trajectory of basal keratinocyte differentiation (Fig. 4a, Supplementary Data 3). SoptSC unbiasedly reconstructed a putative BAS-SPN-GRN keratinocyte differentiation trajectory (Fig. 4b, c). As expected, SoptSC placed basal keratinocytes expressing *KRT14* at the beginning, with average expression levels of *KRT14* declining towards the trajectory terminus, whereas cells expressing the terminal differentiation gene *TGM3* displaying heightened average expression levels towards the trajectory terminus (Supplementary Fig. 11). An alternative method of inferring pseudotemporal trajectories, diffusion pseudotime, also

predicted a similar trajectory between BAS-SPN-GRN clusters (Supplementary Fig. 12).

Next, we asked if the epithelial communities exist on a continuum or have distinct cellular states. Previous studies have suggested that in silico differentiation potency and plasticity of single cells can be approximated by computing the signaling promiscuity of a cell's transcriptome[46]. We developed a similar algorithm to calculate the Cellular Entropy ($\xi$) of single cells called Cellular Entropy Estimator (CEE)–now a feature that has been incorporated into SoptSC (see "Methods"). CEE allows us to estimate the likelihood a single cell will transition from one cellular state to another. We applied CEE to estimate the Cellular Entropy in BAS, SPN, and GRN communities and represented their transition likelihoods in a Waddington landscape, where a

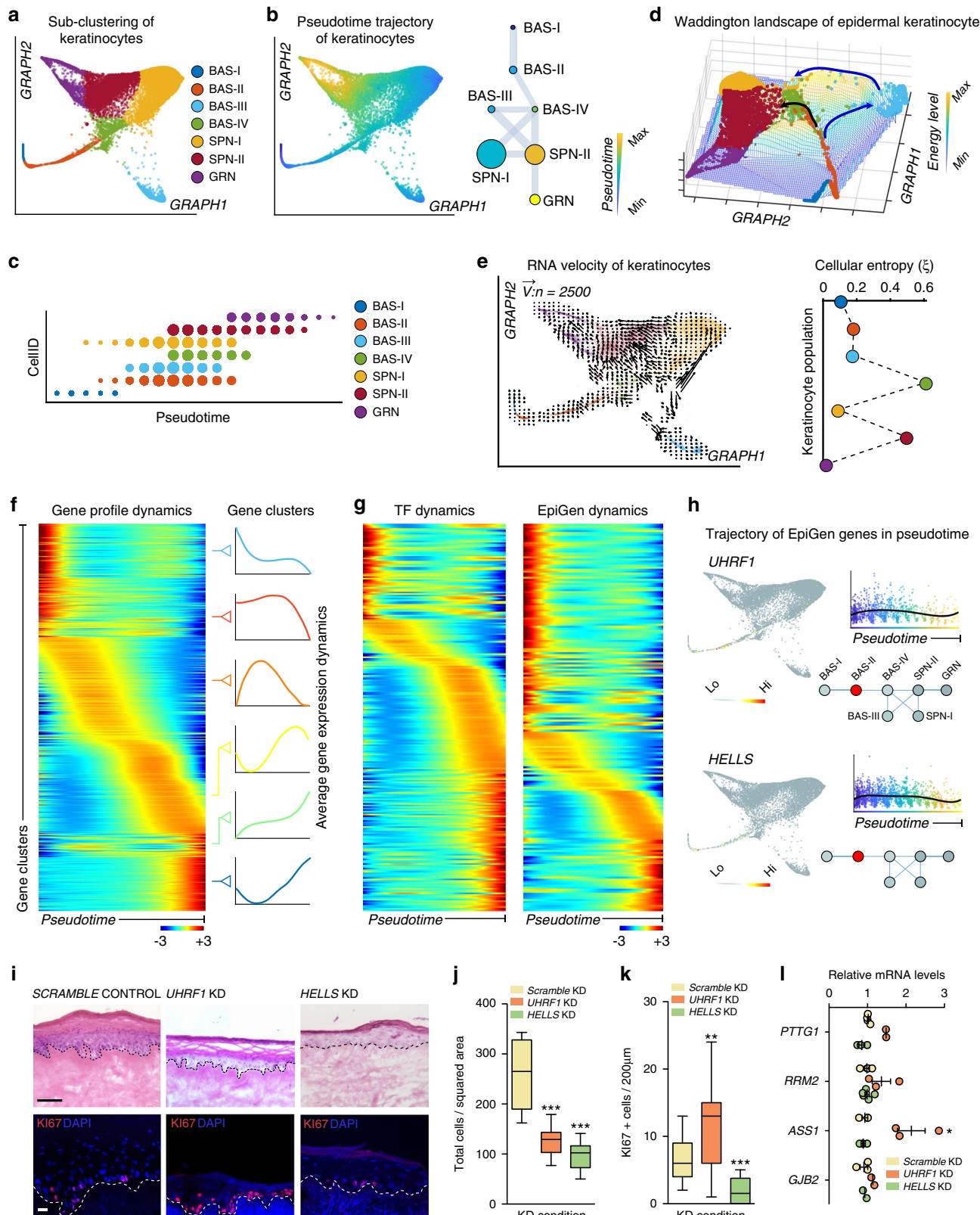

"valley" represents a region of low Cellular Entropy (i.e., low likelihood of transition into a new state) and a "mountain" represents a region of high Cellular Entropy (i.e., high likelihood of transition into a new state) (Fig. 4d). Two transition trajectories emerge that recapitulates the bifurcation in the differentiation trajectory in Fig. 4b, suggesting that BAS-III and

BAS-IV cells can both directly differentiate into spinous cells. Epidermal communities in aggregate displayed distinct entropy values, with BAS-IV having the highest probability of transitioning to a new state ($\xi_{BAS-IV} = \sim 0.60$). As RNA velocity can estimate the future state of cells by analyzing spliced and unspliced variants of mRNA in single cell data[47], we reasoned that

**Fig. 4 Keratinocyte differentiation trajectories highlight epigenetic modifiers in epidermal homeostasis. a** Epidermal keratinocyte subclustering using SoptSC. **b** Pseudotime inference of epidermal keratinocytes. Cell lineage inference displayed on the right. Edge weights denote probability of transition to each cluster. Dot size denotes number of cells. **c** Cell ID vs. pseudotime. Dot size denotes number of cells in pseudotime score. **d** Cellular Entropy ($\xi$) of epidermal keratinocytes plotted in a Waddington landscape. Clusters color-coded as in A. Blue and black arrows show predicted low energy paths along differentiation. **e** RNA velocity of epidermal keratinocytes using 2500 vectors. Quantification of Cellular Entropy displayed on the right. Cells or clusters color-coded as in A. **f** Rolling wave plot showing pseudotime-dependent gene expression dynamics. Average gene expression dynamics of each cluster pattern shown on the right and independently color-coded. **g** Rolling wave plots showing pseudotime-dependent gene expression dynamics of transcription factors (TF) and epigenetic modifiers (EpiGen). **h** Feature plots showing expression of *UHFR1* and *HELLS* and their expression along pseudotime. Expression along epidermal cell lineage displayed at the bottom. **i** Knock-down (KD) of *UHFR1* and *HELLS*, with a scramble control, in human skin equivalent organoids grown for 16 days. Hematoxylin and eosin (H&E) staining shown at top. Scale bar 100 μm. KI67 (red) and DAPI staining shown at the bottom. Scale bar 50 μm. Dotted lines denote the position of the basement membrane. $n = 6$ experiments each condition. **j** Quantification of the total cells per squared area (***$p < 0.0001$; $n = 12$ areas [*HELLS*] or 13 areas [*UHRF1*]), (**k**) KI67 + cells per 200 μm area along the basement membrane (**$p = 0.0012$; ***$p < 0.0001$; $n = 12$ areas [*HELLS*] or 19 areas [*UHRF1*]), and (**l**) qPCR of the respective transcripts for each KD condition (*$p < 0.0292$; $n = 2–4$ experiments each condition). For whisker and box plots: box represents 25th to 75th percentiles, whiskers represent minimum and maximum data points, and bar represents mean. For scatter plot: data are presented as mean values ± SEM. Significance was determined by unpaired two-tailed $t$ test.

combining Cellular Entropy with RNA velocity would predict the transition potency and directionality of a cell or group of cells (i.e., the likelihood of transitioning and its transition directionality). Indeed, we observed that BAS-III and BAS-IV displayed larger velocity vectors pointing toward the spinous clusters (Fig. 4e; Supplementary Fig. 13). SPN-II, having a high entropy value ($\xi_{SPN-II} = \sim 0.5$), also displayed refined velocity vectors toward GRN. *RRM2*- and *PTTG1*-positive cells had low Cellular Entropy values ($\xi_{BAS-I} = \sim 0.1; \xi_{BAS-II} = \sim 0.2$) and lacked refined velocity vectors, suggesting that these cells may represent a steady state.

**BAS-II genes HELLS and UHRF1 affect epidermal homeostasis**. To determine genes that changed along our modeled pseudotime trajectory, we identified pseudotime-dependent gene expression changes and discovered 700 DEGs along the putative BAS-SPN-GRN differentiation trajectory (Fig. 4f and Supplementary Data 4). These DEGs segregated differentially into six distinct clusters according to their average gene expression dynamics along pseudotime. We focused our attention on transcription factors (TFs) and epigenetic modifiers (EpiGens), given their roles in controlling cell states during differentiation of keratinocytes[48]. Using this approach, we identified TFs previously implicated in epidermal homeostasis and differentiation, including *MAFB*, *TP63*, *CEBPA*/B, *KLF4*, *GRHL3*, *GATA3*, and *OVOL1* (Fig. 4g, Supplementary Fig. 11, and Supplementary Data 4). EpiGens included factors involved in DNA methylation such as *DNMT1* and its co-factor *UHRF1*, and covalent histone modifications such as *KDM6B* and *HDAC1/2*. We also identified Polycomb component members, including *EZH1/2*, *JMJ*, and *CBX4*; the ATP-dependent chromatin remodelers *SMARCA4* and *CDH4*; and the higher-order chromatin remodeler *SATB1* (Fig. 4g, Supplementary Fig. 14, and Supplementary Data 4).

Our cell-cell signaling and pseudotime analysis identified the WNT target gene *UHRF1*[49] with heightened expression at the beginning and decreased expression along the pseudotime trajectory. *UHRF1*, along with *HELLS*, are also target genes downstream of the Retinoblastoma (RB) pathway, which is mediated by E2F transcription factors, and have been implicated in recruitment of DNMT1 and DNMT3, respectively[50,51]. HELLS is a SNF2-like helicase, known for its role in silencing chromatin regions via interaction with DNA methyltransferases[52] and functions during development and senescence[53]. Although their role in epidermal homeostasis in unclear, *HELLS* and *UHRF1* are both differentially expressed in the transitional basal BAS-II community (Fig. 4h). To assess the role of *HELLS* and *UHRF1* in epidermal homeostasis, we knocked down (KD) either transcripts in primary human neonatal keratinocytes using viral transduction

and then seeded the genetically modified primary keratinocytes on top of devitalized human dermis to generate human skin equivalents (Supplementary Fig. 15). Histopathological assessment of *HELLS* KD organotypic cultures show decreased epidermal thickness, a significant reduction in total numbers of cells per squared area, and a significant decrease in KI67-positive cycling cells compared with *SCRAMBLE* control (Fig. 4i–k). *UHRF1* KD organotypic cultures also show decreased epidermal thickness, reduced total number of cells, and concomitant increase in KI67-positive cells (Fig. 4i–k). The *UHRF1* KD phenotype is similar to *DNMT1* KD organotypic skin cultures, where epidermal thickness is reduced and an increase in G2/M phase cells is seen at the expense of S phase[54]. BAS-I marker *PTTG1* mRNA levels are significantly increased in *UHRF1* KD keratinocytes, with BAS-II marker *RRM2* trending upwards, recapitulating the increase in cycling cells seen in the *UHRF1* KD organotypic cultures (Fig. 4l). BAS-III marker *ASS1* is also significantly increased in *UHRF1* KD keratinocytes. However, no significant change in any of the BAS cluster markers are seen upon *HELLS* KD (Fig. 4l). Taken together, our functional experiments identify the importance of the epigenetic modifiers *HELLS* and *UHRF1* for the transitional basal cells to regulate human epidermal homeostasis.

**Hierarchically ordered stem cells promote differentiation**. Previous studies have suggested functional heterogeneity within the IFE that has led to four distinct models of how this compartment is formed: (1) a single committed progenitor population that directly self-renews or differentiates[6,7], one SC population that gives rise to (2) TA cells or (3) committed progenitors that directly differentiate[5,8], or (4) two SC populations that regenerate at different rates[9]. To assess functional heterogeneity in the basal compartment and address which model our data supports, we subclustered the four main basal populations (Fig. 5a–c and Supplementary Data 5). Subclustering the four main basal populations provided better resolution of cells expressing differentiation markers and thus likely transitioning into the differentiation state, whereas subclustering *KRT14*-high expressing cells resulted in sub-sampling the existing basal subpopulations with an absence of differentiating cells (Supplementary Fig. 16). After subclustering, 7 subpopulations emerged that split the BAS-II, BAS-III, and BAS-IV clusters (Fig. 5c). The transitional BAS-I cluster remained intact, further supporting its low Cellular Entropy value and robust steady state (Fig. 4e). Pseudotime analysis indicates a bifurcated differentiation trajectory that is supported by RNA velocity (Fig. 5d, e; Supplementary Fig. 13). The bifurcation in the pseudotime analysis seems to be the result of cellular trajectories going from kBAS-II and kBAS-III to kBAS-

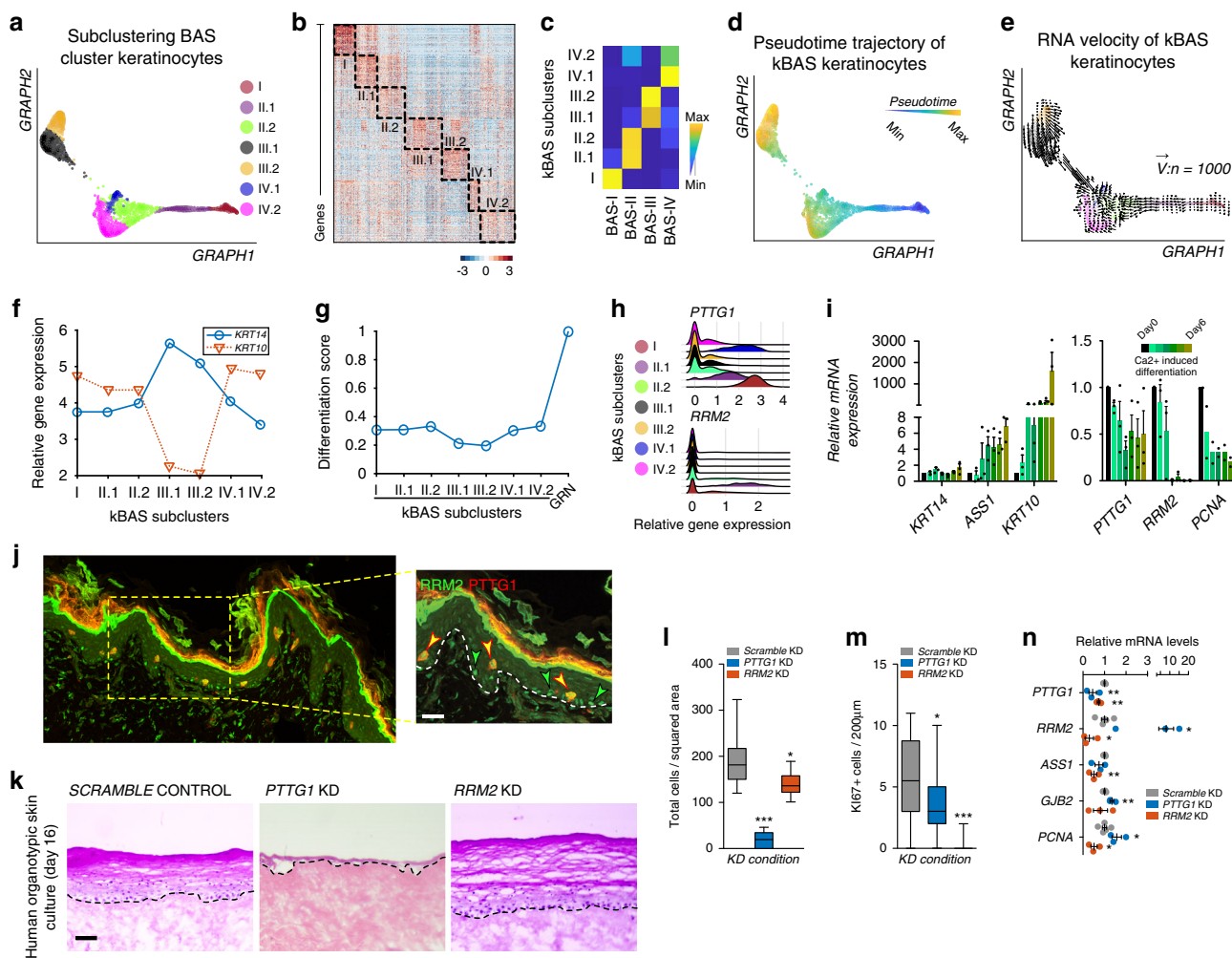

**Fig. 5 Subclustering basal stem cells point to the BAS-I cluster as essential to epidermal homeostasis. a** Subclustering basal clusters (BAS-I – BAS-IV) using SoptSC. **b** Heatmap showing top 100 differentially expressed genes per cluster. **c** Correlation between overlapping differentially expressed genes from basal clusters and subclustering of basal clusters (kBAS) in A. **d** Pseudotime inference of kBAS keratinocytes. **e** RNA velocity of kBAS keratinocytes using 1000 vectors. Cells color-coded as in A. **f** Relative gene expression of *KRT14* and *KRT10* across the kBAS subclusters. **g** Differentiation score of kBAS subclusters using average gene expression of GRN genes. **h** Violin plot of relative gene expression of *PTTG1* and *RRM2* across kBAS subclusters. **i** Relative mRNA expression of indicated genes along $Ca^{2+}$-induced differentiation of primary human keratinocytes. $n = 3$ experiments each time point ($n = 2$ for *PCNA*). **j** Coimmunostaining of PTTG1 (red) and RRM2 (green) in human neonatal skin. Green arrowheads highlight RRM2-only expressing cells; yellow arrowheads point to PTTG1-RRM2 double-positive cells. White dotted line denotes basement membrane. Scale bar 50 μm. $n = 3$. **k** Knock-down (KD) of *PTTG1* and *RRM2*, with a scramble control, in human skin equivalent organoids grown for 16 days. Hematoxylin and eosin (H&E) staining shown. Dotted lines denote basement membrane. Scale bar 100 μm. $n = 4$ experiments each condition. **l** Quantification of total cells per squared area (*$p = 0.0104$; ***$p < 0.0001$; $n = 12$ areas each [*PTTG1* and *RRM2* KD]), (**m**) KI67 + cells per 200 μm area along the basement membrane (*$p = 0.0123$; ***$p < 0.0001$; $n = 26$ areas [*PTTG1*] or $n = 25$ areas [*RRM2*]), and (**n**) qPCR of the respective transcripts for each KD condition (*PTTG1*: $p = 0.0056$, 0.0018; *RRM2*: $p = 0.0485$, 0.025; *ASS1*: $p = 0.0025$; *GJB2*: $p = 0.0037$; *PCNA*: $p = 0.0339$, 0.0295; $n = 3$ experiments each [*PTTG1* and *RRM2* KD]). For whisker and box plots: box represents 25th to 75th percentiles, whiskers represent minimum and maximum data points, and bar represents mean. For bar graph: error bar represents SEM. For scatter plot: data are presented as mean values ± SEM. Significance was determined by unpaired two-tailed *t* test.

IV subpopulations as seen in the RNA velocity plot (Fig. 5e). Tracking *KRT14/KRT10* gene expression in the kBAS subpopulations as a proxy for differentiation status shows a gradual decrease in *KRT14* gene expression from kBAS-III to kBAS-IV with a concomitant increase *KRT10* expression, further supporting a linear temporal trajectory towards differentiation that occurs before commitment to differentiation (Fig. 5f). When pairing the *KRT14/KRT10* expression with a differentiation score that tracks expression of GRN-specific genes, we observe commitment to differentiation occurring in the kBAS-IV subpopulation at the end of pseudotime (Fig. 5g).

Incidentally, the kBAS-IV.2 cluster most resembles the *Ivl-CreER+* committed progenitors in murine tail epidermis[8], with

their *Krt14-CreER+* SCs most resembling the kBAS-I cluster (Supplementary Fig. 17). In addition, nonlabel retaining SCs with high proliferative capacity in murine dorsal back epidermis from *Krt5-tTa*; *pTRE-H2B-GFP*; *Krt14-CreER*; *Rosa-tdTomato* mice[9] overlap with kBAS-I and *Krt14-CreER+* SC datasets, whereas the label retaining SCs with low proliferation capacity most resemble the KBAS-III.1 subpopulation that is also enriched for the basal SC marker *ITGB1* (Supplementary Fig. 17). Although mouse tail epidermis contain interscale (orthokeratotic; most similar to dorsal back epidermis) and scale (parakeratotic; lack of granular layer and retention of nuclei in cornified layers) IFE each with their own proliferative capacity, clones originating in either region can cross boundaries[55], suggesting that they arise from

similar basal populations that are differentially regulated. In addition, the segregation of label retaining and nonlabel retaining SCs in dorsal back epidermis suggests structural similarities between back and tail skin[9]. These data, coupled with pseudotime and lineage inference indicating that the basal populations are hierarchically ordered (Fig. 4b) and RNA velocity indicating BAS-III and BAS-IV can both contribute to spinous cells (Fig. 4e), suggest that multiple SC populations contribute to differentiation and agree with models presented by the Blanpain and Tumbar groups that describe multiple SC pools with different proliferation capacities[8,9].

We further profiled the kBAS subpopulations by assessing gene expression markers that encompass both transitional basal populations (BAS-I = kBAS-I and BAS-II = kBAS-II) and encompass the *Krt14-CreER* + SCs and nonlabel retaining cells with high proliferative capacity. kBAS-I can be spatially defined by expression of *PTTG1* (Fig. 5h), a proto-oncogene that is involved in controlling keratinocyte proliferation, early stages of differentiation, cell growth, and is overexpressed in psoriasis[56,57]. kBAS-II can be spatially defined by expression of *RRM2* (Fig. 5h), which controls the biogenesis of dNTPs and is overexpressed in skin cancer[58]. Expression of *PTTG1*, *RRM2*, and fellow kBAS-II gene *PCNA* decreased in keratinocytes during $Ca^{2+}$-induced differentiation (Fig. 5i). In addition, co-immunofluorescence staining of the proteins showed RRM2-only expressing cells largely restricted to the basal layer, whereas double-positive RRM2 and PTTG1 cells occupying both basal and suprabasal positions, further reinforcing our observation that these genes are expressed in undifferentiated keratinocytes (Figs. 2m and 5j). To determine the role of *PTTG1* in epidermal maintenance, we performed *PTTG1* KD in primary human neonatal keratinocytes using viral transduction and then seeded the genetically modified primary keratinocytes on top of devitalized human dermis to generate human skin equivalents (Supplementary Fig. 15). *PTTG1* KD led to a severe epidermal phenotype, primarily characterized by disruption of the basal layer and differentiation program compared with scramble shRNA controls, indicating its essential role in basal SC maintenance and epidermal homeostasis (Fig. 5k, l). On the other hand, *RRM2* KD displayed a less severe but still significant phenotype in epidermal homeostasis in human skin equivalents (Fig. 5k, l). *PTTG1* and *RRM2* KD organotypic cultures both show a significant decrease in KI67-positive cycling cells compared with *SCRAMBLE* control, with *PTTG1* KD cultures largely devoid of KI67-positive cells (Fig. 5m). BAS-II markers *RRM2 and PCNA* mRNA levels are significantly increased in *PTTG1* KD keratinocytes, with BAS-IV marker *GJB2* significantly increased (Fig. 5n). Although *RRM2* KD organotypic cultures show a less severe epidermal thickness phenotype compared with *PTTG1* KD, *RRM2* KD cultures do show a significant decrease in most of the tested BAS markers (Fig. 5n).

## Discussion
Functional heterogeneity in human IFE has been largely unexplored compared with their murine counterparts. We made the discovery of at least four basal SC populations in human neonatal epidermis using scRNA-seq. Each population is spatially distinct, with BAS-III cells occupying space between rete ridges and BAS-IV residing at the tips or bottom of the rete ridges, whereas the BAS-I and BAS-II populations showing sparse and heterogenous distribution throughout the basal and suprabasal layers. Analyzing cell-cell communication, gene profile dynamics, and genetic loss-of-function experiments indicate that the WNT target gene and epigenetic modifiers UHRF1 and HELLS and the proto-oncogene PTTG1 are essential for epidermal homeostasis. Finally, our results provide clarity in the various models of IFE

homeostasis and suggests that multiple SC pools with different proliferation capacities contribute to differentiation and epidermal homeostasis in humans.

The heterogeneity of basal SCs in human IFE should not be surprising given that scRNA-seq studies have found robust heterogeneity in nearly all of the profiled tissues. However, KRT14 staining and use of the *Krt14-Cre* murine reporter lines as a proxy for the basal SC population lend to the inaccurate assumption of homogeneity in the epidermal basal layer. Use of mouse Cre reporter lines to untangle this homogeneity has led to models of IFE homeostasis that suggest two potentially distinct SC populations[8,9], while others maintain one SC population[6,7]. Instead of the one or two basal SC populations from other models, our results indicate at least four spatially distinct basal populations exist in human neonatal epidermis. Intriguing hints in human studies point to this heterogeneity, where ITGB1 expression seemingly sits between rete ridges[59], an observation we confirm with our ASS1 and COL17A1 immunostaining (Fig. 2f). In fact, this population also appears to express *ITGA6* and *TP63*, suggesting it is the classically defined human basal SC population (Supplementary Fig. 17). In addition, ALP staining at the base of rete ridges[27] suggests another potentially unique basal population we confirm with our GJB2 and KRT19 immunostaining (Fig. 2g).

Our data indicates a linear hierarchy of differentiation beginning at the *Krt14-CreER* + /nonlabel retaining SC population (BAS-I), transitioning to the label retaining SC and *ITGB1*-high populations (BAS-II and BAS-III), and committing to differentiate in the *Ivl-CreER* + committed progenitor population (BAS-IV). Once committed, the spinous cell populations appear to mature to terminally differentiated granular keratinocytes on a continuum where no gene clearly marks the spinous layer despite their respectively distinct morphologies. In practice, however, both the BAS-III and BAS-IV clusters appear to contribute to the spinous cell populations. The spatial positioning of BAS-III and BAS-IV also suggest they both give rise to differentiated progeny as we observe KRT10 immunostaining beginning in the second layer along the rete ridges and flat portions of IFE (Fig. 2e). The spatial positioning of the transitional basal populations (BAS-I and BAS-II), so-called because of their unusual spatial position where a significant percentage of the cells remain attached to the underlying basement membrane while their cell body is in a suprabasal position, suggest they may be delaminating from the basal layer. However, these basal cells are still attached to the basement membrane and may remain in the basal layer over time. Live imaging of these fate transitions would help shed light on their movements.

Why does interfollicular epidermal homeostasis need multiple SC populations? Skin appendages originate from the epidermal layer and it is possible that the epidermis actively responds to inductive signals even in the adult. For instance, normal epithelial cells are required for mutant epithelial cells resulting in aberrant growths to regress and are either eliminated or converted into functional skin appendages[60], suggesting that basal cell heterogeneity may be protective against harmful insults. In addition, basal SC self-renewal is not a constitutive driver of epidermal homeostasis; rather, cells need the plasticity to respond to their environment and coordinate divisions with stochastic loss of neighboring cells to maintain homeostasis[61]. Our results suggest that specific genes within the transitional basal populations (BAS-I and BAS-II) are essential for epidermal homeostasis in human skin equivalents. Disruption of *PTTG1* expression (BAS-I marker gene) results in severe loss of epidermal stratification, resembling simple epithelium and displaying a more severe phenotype than was previously reported[56]. And disruption of either *HELLS* or *UHRF1* expression (BAS-II marker genes) results in suppression of epidermal homeostasis and a thinner epidermis. HELLS is

thought to recruit UHRF1 and DNMT1 to sites of methylated chromatin to facilitate demethylation and all three show strong similarities when disrupted in human skin equivalents (HELLS and UHRF1: Fig. 4i; DNMT1[54]), in stark contrast to the mouse phenotype where *Krt14-Cre*-mediated loss of *Dnmt1* results in an increase in IFE proliferation and aberrant differentiation[62]. Differences in the frequency of KI67 + cells in *HELLS* and *UHRF1* KD skin equivalents suggest distinct differences between their functional roles in human epidermis despite their overlapping role in DNA demethylation.

Overall, our findings illustrate the dynamic nature of basal keratinocytes in human neonatal epidermis. The signaling and lineage relationships between the basal SC populations and differentiated keratinocytes warrants further characterization of the factors influencing fate plasticity and may help uncover the similarities and differences between vertebrate epidermal homeostasis and disease.

## Methods

**Ethics statements**. Human clinical studies were approved by the Ethics Committee of the University of California, Irvine. All human studies were performed in strict adherence to the Institutional Review Board (IRB) guidelines of the University of California, Irvine (2009-7083). We have obtained informed consent from all participants.

**Histology and immunohistochemistry**. Frozen tissue sections (10 μm) were fixed with 4% PFA in PBS for 15 min. Following fixation, tissue sections were stained with Hematoxylin and Eosin following standard procedures. Sections were stained with Gill's III (Fisher Scientific; 22050203) for 5 min and Eosin-Y (Fisher Scientific; 22050197) for 1 min. Tissue sections were visualized under a light microscope under 10x objective lens after mounting with Permount mounting media (Fisher Scientific; SP15-100). For immunostaining, tissue sections were fixed with 4% PFA in PBS for 15 min. 10% BSA in PBS was used for blocking. Following blocking, 5% BSA and 0.1% Triton X-100 in PBS was used for permeabilization. The following antibodies were used: chicken anti-KRT14 (1:500; BioLegend; SIG-3476), mouse anti-KRT10 (1:500; Dako; M7002), rabbit anti-KI67 (1:500; Abcam; ab15580), rabbit anti-PTTG1 (1:100; Sigma-Aldrich; HPA008890), mouse anti-ASS1 (1:100; Santa Cruz; sc-365475), rabbit anti-COL17A1 (1:100; One World Labs; ap9099c), rabbit anti-COL7A1 (1:500; abcam; ab93350), mouse anti-COL7A1 (1:100; Santa Cruz; sc-33710), rabbit anti-GJB2 (1:250; ThermoFisher; 51-2800), rabbit anti-KRT19 (1:250; Cell signaling; 13092), mouse anti-SPINK5 (1:100; Santa Cruz; sc-137109), mouse anti-CALML5 (1:100; Santa Cruz; sc-393637), mouse anti-CDC20 (1:100; Santa Cruz; sc-13162), mouse anti-RRM2 (1:100; Santa Cruz; sc-398294, sc-376973), mouse anti-PCLAF (1:100; Santa Cruz; sc-390515), mouse anti-MLANA (1:100; Santa Cruz; sc-20032), and mouse anti-CD74 (1:100; Santa Cruz; sc-6262). Secondary antibodies included Alexa Fluor 488 (1:500; Jackson ImmunoResearch; 715-545-150, 711-545-152) and Cy3 AffiniPure (1:500; Jackson ImmunoResearch; 711-165-152, 111-165-003). Slides were mounted with Prolong Diamond Antifade Mount containing DAPI (Molecular Probes). Confocal images were acquired at room temperature on a Zeiss LSM700 laser scanning microscope with Plan-Apochromat 20x objective or 40x and 63x oil immersion objectives. Images were arranged with ImageJ, Affinity Photo, and Affinity Designer.

**Protein immunoblotting**. Cells were lysed with 2X SDS sample buffer (100 mM Tris HCl 6.8, 200 nM DTT, 4% SDS, 0.2% bromophenol blue, and 20% glycerol) and boiled at 100 °C for 10 min. Samples were resolved on a 12.5% polyacrylamide gel and transferred to nitrocellulose membrane by a wet transfer apparatus. Membranes were blocked with 5% milk in TBS with 0.05% Tween-20 before sequential addition of primary and secondary antibodies. Membranes were imaged using Alexa Fluor secondary antibodies and the LI-COR Odyssey imaging system. Secondary antibodies included Alexa Fluor 680 (1:2000; Jackson ImmunoResearch; 715-625-150, 711-625-152) and Alexa Fluor 790 (1:2000; Jackson ImmunoResearch; 711-655-152).

**Lentiviral knockdown**. Either pSicoR or pGIPZ vectors were used for lentiviral knockdown of specific genes. pSicoR was a gift from Tyler Jacks (Addgene; 11579). For pSicoR, shRNA were designed using pSicoligomaker 3.0 (Ventura Lab) and cloned using InFusion HD Cloning Plus Kit (Takara, 638911). The following sequences were used: PTTG1 5′-GATGATGCCTATCCAGAAATTCAAGAGA TTTCTGGATAGGCATCATC-3′; RRM2 5′-GCACTCTAATGAAGCAATATT CAAGAGATATTGCTTCATTAGAGTGC-3′. Lentiviral pGIPZ vectors contained shRNAs to *UHRF1* (5′-TGACATTGCGCACCACCCT-3′) and *HELLS* (5′-ACAGGCTGATGTGTACTTAACC-3′)[63] (Dharmacon). Transduced cells were selected via Puromycin (ThermoFisher; 50464455). Knockdown efficiency was determined by protein levels on Western Blot or by quantitative RT-PCR where

fold change in mRNA expression was measured using $\Delta\Delta C_T$ analysis with *GAPDH* as an internal control. The following qPCR primers were used: PTTG1 forward 5′-CCCTTGAGTGGAGTGCCTCT-3′, reverse 5′-CACAGCAAACAGGTGGCAA T-3′; RRM2 forward 5′-GCAGCAAGCGATGGCATAGT-3′, reverse 5′-GGGCTT CTGTAATCTGAACTTC-3′; PCNA forward 5′-CGACACCTACCGCTGCGAC C-3′, reverse 5′-TAGCGCCAAGGTATCCGGCGT-3′; and GAPDH forward 5′-GC ACCGTCAAGGCTGAGAAC-3′, reverse 5′-TGGTGAAGACGCCAGTGGA-3′.

**Preparation of devitalized dermis**. Cadaver human skin was acquired from the New York Firefighters Skin Bank (New York, New York, USA). Upon arrival at UC Irvine, the skin was allowed to thaw in a biosafety hood. Skin was then placed into PBS supplemented with 4X Pen/Strep, shaken vigorously for 5 min, and transferred to PBS supplemented with 4X Pen/Strep. This step was repeated two additional times. The skin was then placed into a 37 °C incubator for 2 weeks. The epidermis was removed from the dermis using using sterile watchmaker forceps. The dermis was washed 3 times in PBS supplemented with 4X Pen/Strep with vigorous shaking. The dermis was then stored in PBS supplemented with 4X Pen/Strep at 4 °C until needed.

**Primary cell isolation**. Discarded and deidentified neonatal foreskins were collected during routine circumcision from UC Irvine Medical Center (Orange, CA, US). The samples were either processed for histological staining, scRNA-seq, or primary culture. No personal information was collected for this study. For primary cell isolation, fat from discarded and deidentified neonatal foreskins were removed using forceps and scissors and incubated with dispase epidermis side up for 2 h at 37 °C. The epidermis was peeled from the dermis, cut into fine pieces, and incubated in 0.25% Trypsin-EDTA for 15 min at 37 °C and quenched with chelated FBS. Cells were passed through a 40 μm filter, centrifuged at 1500 rpm for 5 min, and the pellet resuspended in Keratinocyte Serum Free Media supplemented with Epidermal Growth Factor 1-53 and Bovine Pituitary Extract (Life Technologies; 17005042). Cells were either live/dead sorted using SYTOX Blue Dead Cell Stain (ThermoFisher; S34857) for scRNA-seq or incubated at 37 °C for culture.

**Cell sorting**. Following isolation, cells were resuspended in PBS free of $Ca^{2+}$ and $Mg^{2+}$ and 1% BSA and stained with SYTOX Blue Dead Cell Stain (ThermoFisher; S34857). Samples were bulk sorted at 4 °C on a BD FACSAria Fusion using a 100 μm nozzle (20 PSI) at a flow rate of 2.0 with a maximum threshold of 3000 events/s. Following exclusion of debris and singlet/doublet discrimination, cells were gated on viability for downstream scRNA-seq.

**Human organotypic skin culture**. Primary human keratinocytes were cultured in Keratinocyte Serum Free Media supplemented with Epidermal Growth Factor 1-53 and Bovine Pituitary Extract (Life Technologies; 17005042). For generating organotypic skin cultures, ~500 K control or knockdown cells were seeded on devitalized human dermis and raised to an air/liquid interface in order to induce differentiation and stratification over the indicated number of days with culture changes every 2 days[64].

**Droplet-enabled single cell RNA-sequencing and processing**. Cell counting, suspension, GEM generation, barcoding, post GEM-RT cleanup, cDNA amplification, library preparation, quality control, and sequencing was performed at the Genomics High Throughput Sequencing Facility at the University of California, Irvine. Transcripts were mapped to the human reference genome (GRCh38) using Cell Ranger Version 2.1.0.

**Quality control metrics post-Cell Ranger assessment**. For downstream analyses, we kept cells which met the following filtering criteria per biological replicate per condition: >200 and <5000 genes/cell, and <10% mitochondrial gene expression. Genes that were expressed in less than 3 cells were excluded. Data were normalized with a scale factor of 10,000.

**Analysis and visualization of processed sequencing data**. Seurat and SoptSC[26] were implemented for analysis of scRNA-seq data in this study. Seurat was performed in R (version 2.2) and was applied to all the datasets in this study. To select highly variable genes (HVGs) for initial clustering of cells, we performed Principal Component Analysis on the scaled data for all genes included in the previous step. For clustering, we used the function *FindClusters* that implements Shared Nearest Neighbor modularity optimization-based clustering algorithm on 20 PC components with resolution 0.6. Nonlinear dimensionality reduction methods, namely tSNE and UMAP, were applied to the scaled matrix for visualization of cells in two-dimensional space using first 10 PC components. The marker genes for every cluster compared with all remaining cells were identified using the *FindAllMarkers* function. For each cluster, genes were selected such that they were expressed in at least 25% of cells with at least 0.25-fold difference.

SoptSC was performed in MATLAB (version 2017b). In order to make both methods comparable for data clustering and downstream analysis, we used Seurat for quality control and normalization and used the resulting data matrices as an input for SoptSC. SoptSC selected 3000 HVGs for the downstream analysis where

the genes with highest loading in the first $k$ principal components were selected ($k$ is the index where the largest gap of the principal component variances occurs). A cell-to-cell similarity matrix preserving both local and global structure of the single-cell data was learned via SoptSC by solving an optimization problem. SoptSC then perform clustering, marker gene identification, lineage inference through a single mathematical framework. Visualization of single cell data in low-dimensional space was implemented by applying Elastic Embedding (EE), a nonlinear dimensionality reduction technique, to the cell-to-cell similarity matrix. The number of common marker genes between clusters among the top 500 marker genes for each cluster identified from two pipelines was compared with assess clustering correlations between Seurat and SoptSC.

**Pseudotime and lineage inference.** Melanocytes and Langerhans cells were regressed out for lineage and pseudotime analysis. The resulting data post-melanocyte and Langerhans cell removal was reclustered using SoptSC, where the number of HVGs was set to 10,000. Pseudotime and lineage analysis were performed using SoptSC. Briefly, pseudotime was calculated as the shortest path distance between cells and root cell on the cell-to-cell graph constructed based on the similarity matrix. Root cell was identified by the user in SoptSC. Visualization of the cell trajectories was obtained from EE with similarity matrix taken as an input. Cell states were visualized using abstract lineage trees. Lineage trees are obtained by computing the minimum spanning tree of the cluster-to-cluster graph based on the shortest path distance between cells. Pseudotime was projected on the lineage tree such that the order of each state (cluster) was defined as the average distances between cells within the state and the root cell.

Diffusion pseudotime (DPT), which measures transitions between cells using diffusion-like random walks, was also used to infer pseudotime[65]. The matlab-version of DPT was used. We took gene expression matrices with overall 700 DEGs (top 100—DEGs for each cluster) identified from SoptSC as input for DPT. The root cell for DPT was selected from the BAS-I cluster.

**RNA velocity.** RNA velocity was estimated based on the spliced and unspliced transcript reads from the single-cell data[47]. We followed the standard process of the velocyto pipeline to generate the spliced and unspliced matrices by applying *velocyto.py* to the data from the Cell Ranger output (outs) folder. We remove melanocytes and Langerhans cells for the velocity analysis and only epithelial cells included in pseudotime were used to calculate velocity vectors. RNA velocity was estimated using a gene-relative model with 25 nearest neighbors and then the velocity fields were projected onto the EE space produced by SoptSC. We set the parameter $n$ sight as 2500, which defines the size of the neighborhood used for projecting velocity. We also applied the RNA velocity analysis for basal cells using the similar procedure where the parameter $n$ was set as 500. Default settings were used for the rest of the parameters.

**Probabilistic cell-cell signaling networks.** Cell-cell communication was determined in SoptSC via signaling networks for JAK/STAT, NOTCH, TGF-β, and WNT signaling pathways. In such networks, the probability between two cells is quantified by interactions between specific ligand-receptor pairs and their downstream target genes[26]. Prior calculation of the cell-cell interaction probability, we performed nature log normalization of the input count matrix. The lists of ligand-receptor pairs were determined before[29,30] and with a survey of current literature. The target genes for each pathway were identified from NetPath[28]. Circos plots (R Studio Version)[66] are utilized as a visualization of cell-cell signaling networks. Edges between cells represent an interaction between them, and the width of each edge represents the interaction probability. Arrows start from ligand and points to receptor. We set a threshold (φ = 0.1) such that the probability is restricted to zero if its value is less than φ. Cluster-level communications were naturally employed by calculating the average value of probabilities between cell-cell interactions from two clusters. In the cluster-to-cluster signaling network plots for each specific pathway, the width of the edge represents the probability value between clusters. Arrows start from ligand and points to receptor. Ligand-receptor pairs and their targets genes among all inferred cell populations are also presented as heatmaps. For each gene, the average expression within each population was calculated.

In order to identify similar patterns for ligand-receptor pairs with respect to cluster-to-cluster interactions, we clustered the ligand-receptor pairs by comparing the similarity between the cluster-to-cluster interactions associated with each specific pair. The clustering procedure includes two steps: projecting the ligand-receptor pairs into two-dimensional space via tSNE in MATLAB and then applying a hierarchical clustering method to the projected data. Ligand-receptor pairs with similar cluster-to-cluster interactions were grouped together and a circos plot of cluster-level interaction was produced for each group of pairs by taking the average of all the interactions within the group.

To determine consistency scores for signaling interactions between libraries, we sum all the differences between the overlapped cluster-cluster interaction probabilities in the reference library (library 3) and target library and divided by the number of nonzero cluster-cluster interactions in the reference library. The consistency score is defined as one minus the above value. Higher scores indicate better consistency between the two libraries.

**Gene ontology analysis.** Top 100 DEGs from each basal cluster were used for gene ontology and pathway analyses using Enrichr[67].

**Cellular entropy estimation.** Cellular Entropy ($\xi$) measures the likelihood that a cell will transition to a new state (i.e., from one cluster to another). Lower entropy values indicate that the cell remains in a steady state, while higher entropy values imply the cell inherits multiple state properties and is more likely to transition to a new state. Via the non-negative matric factorization step in SoptSC, the probability of each cell assigned to each cluster is calculated (i.e., $P_{i,j}$ for cell $i$ and cluster $j$). The entropy for each cell is then defined as:

$$\xi_i = -\sum_j^K P_{i,j} \log(P_{i,j}), \qquad (1)$$

where $K$ represents the number of clusters. To visualize the trajectory of cells and their likelihood of transition along a transition valley, we constructed a Waddington landscape and overlaid the cell states on it. The Waddington landscape is constructed by integrating the low-dimensional representation of our data (via applying EE to similarity matrix) and the cellular entropy estimation for each cell. This feature has been extended to the current methods employed by SoptSC.

## Data availability
The authors declare that all data supporting the findings of this study are available within the article and its supplementary information files or from the corresponding author upon reasonable request.

The datasets generated during the current study have been deposited in the GEO database under accession code GSE147482. Source data are provided with this paper.

## Code availability
Source code to reproduce data analysis is available on github [https://github.com/WangShuxiong/Human_Epi].

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

## Acknowledgements

S.X.A. is supported by the Concern Foundation (CF204525), NIH grant P30AR075047, and start-up funds from UCI. Q.N. is supported by NIH grants U01AR07315, R01GM123731, and P30AR075047, a NSF grant DMS11763272, and a Simons Foundation grant (594598). C.F.G.-J. is supported by UC Irvine Chancellor's ADVANCE Postdoctoral Fellowship Program, NSF-Simons Postdoctoral Fellowship, and a kind gift from the Howard Hughes Medical Institute Hanna H. Gray Postdoctoral Fellowship Program. E.T. and A.R.S. are supported by the GAANN Fellowship. We thank Jennifer Bates and the UCI Institute for Immunology Flow Cytometry Core Facility for help with cell sorting.

## Author contributions

S.X.A. and Q.N. conceived the project; S.X.A. and Q.N. supervised research; M.L.D. generated scRNA-seq libraries; S.W. and M.L.D. analyzed scRNA-seq data; S.W. performed SoptSC and most bioinformatic analyses; M.L.D., E.T., G.G., and A.R.S. performed imaging experiments; M.L.D., E.T., S.C.W., and B.T.T. performed knockdown experiments and analysis; S.W., C.F.G.-J., A.L.M., M.L.D., Q.N., and S.X.A. analyzed and interpreted data; C.A.B. contributed *HELLS* and *UHRF1* KD data and reagents; S.X.A., C.F.G.-J., and S.W. wrote the manuscript. All authors analyzed and discussed the results and commented on the manuscript.

## Competing interests

The authors declare no competing interests.
