## [Peer Review File · Nature Communications]

Reviewers' Comments:

Reviewer #1:

Remarks to the Author:

In this manuscript by Wang et al. the authors utilize Single-Cell-RNA-Sequencing by 10X genomics to explore cellular heterogeneity of human neonatal foreskin epidermis. To my knowledge this is one of the first single-cell-RNA-seq analysis of the epidermis of neonatal human skin. Neonatal human skin is difficult to acquire with foreskin epidermis being the most practical to gather. Furthermore, neonatal mammalian skin is very different than adult tissue. This is because neonatal skin is at a point of generating new skin to grow in conjunction with the organism compared to resting adult skin, making the analysis performed by the authors critically important.

The authors analyzed their single cell data sets with multiple computational pipelines, including; Seurat, Monocle, and SoptSC. These analyses have identified multiple populations of basal-progenitor cells in neonatal epidermis in addition to the differentiated layers such as the spinous keratinocytes. These clusters are interesting but expected to be found in Single-Cell-RNA-seq experiments. Importantly, the authors validate their initial analysis by using novel SoptSC software packages that produces investigate cell-cell interactions in addition to the Cellular Entropy Estimator. This result was highly interesting because it adds an additional dimension to RNA-velocity analysis, which was also performed, to reveal the potential for epidermal progenitor differentiation along a trajectory.

The authors used the multiple biocomputational pipelines to identify new genes (epigenetic-UHRF1/Hells and Pttg1, Rrm2, Ass1, Col17a1 etc) to generate a novel molecular map of the basal layer of the skin. In addition, the authors tested whether the identified molecular factors were critical for the formation of the establishment of skin equivalents in culture, which revealed their importance in the formation of healthy epidermis.

This manuscript is well written, and the data is novel with multiple detailed computational analysis performed in a statistically rigorous manner for Single Cell RNA seq data. The manuscript is of very high quality and is ready for publication.

Reviewer #2:

Remarks to the Author:

The paper by Shuxiong Wang et al. describes a comprehensive single-cell transcriptomics and computational analysis of cellular heterogeneity in human interfollicular epidermis. The authors find stable functional heterogeneity that correlates with spatial heterogeneity and is further corroborated by functional analyses. The authors identify HELLS and PPTG as novel factors involved in epidermal homeostasis and differentiation.

While the paper has great potential, I propose that the following issues are addressed before publication:

Major

- The paper describes droplet-based scRNA-seq of primary cells from five donors. However, many of the presented analyses are based on a single donor. Supplementary Figure 4 indicates some variability in the cell clustering from different donors. For example, the GRN cluster is not detected in all of the donors. In my opinion, it would be important to show that the key findings from analyses of Library 3 are reproducible across donors. These include the expression of key marker genes (highlighted in Figure 2A) and the results of the pseudotime analysis.
- On page 9 the authors state that "SoptSC unbiasedly reconstructed a putative BAS-SPN-GRN keratinocyte differentiation trajectory (Figure 4B)". However, the figure does not clearly show GRN being the end point of the trajectory. Instead, based on 4A-B it seems that some of the GRN cells in fact precede SPN cells in pseudotime. In my opinion, the figure could be improved by addition of a panel visualizing pseudotime vs. cell identities (using the colour scheme from panel A).

Minor

- The cell-cell network inference analysis is rather difficult to interpret from Figure 3. From the heatmaps it would seem that the Basal cell cohort expresses on average the highest levels of genes of each pathway, yet based on the circos plots they are not among the active clusters. Could the authors please explain this discrepancy? Would it be make sense to show the expression of ligands and receptors separately? Finally, in the methods section it is suggested that the circos plots have directionality indicated by arrows. However, the figure is much too small to see such detail.
- Combining entropy and RNA velocity is an elegant and innovative approach for delineating differentiation trajectories. Unfortunately, panel 4E is too small for seeing the direction of the vectors. Please improve this as it is central information. (The same applies also to Fig. 5E)
- P. 18 methods: ". For downstream analyses, we kept cells which met the following filtering criteria per biological replicate per condition: <6000 UMI/cell, and <10% mitochondrial gene expression. " Is this UMI filter correct? From Supplementary Figure 2 it is evident that a significant fraction of cells has more than 6000 UMIs.
- Figure 1C and 1G legends are identical – in the interest of clarity it would be good to point out the different clustering methods used for defining the cohorts
- Figure 1H does not seem to be cited in the text
- In the results section supervised clustering in Seurat is mentioned. But the methods section describes unsupervised clustering. Which one was used in fact?
- Typos: P.20 "circus plots" -> circos plots

Reviewer #3:

Remarks to the Author:

The authors of the submitted manuscript perform single cell RNAseq analysis of neonatal foreskin epidermal keratinocytes. Based on the analysis the authors propose the existence of four distinct populations of basal cells, and provide some evidence for heterogeneity within the basal layer of the epidermis based on expression of markers. However, the authors provide no quantification of their staining nor coevaluation of the proposed markers. The authors go on to infer based simply on RNA expression analysis that paracrine and autocrine signaling are essential for homeostasis without any validating data. They subsequently propose that cells are organized along discrete cellular trajectories with key transition states. Functional analysis of a couple of markers for different markers associated with the basal population forms the basis for proposing that this further support that cells are organized in a cellular hierarchy and that specific epigenetic regulators control progenitor behavior.

Although the dataset could have some interest for the skin community, the manuscript appears somewhat premature, lacking general controls and quantification of the different described phenotypes and expression patterns. Several concerns with both the analyses and the interpretation of the resulting data are outlined below.

Major comments

First and foremost, the authors clearly state that they produced 5 libraries for single cell analyses, however, they have chosen to focus on one library from a single individual. Human samples are heterogenous by nature and focusing the analysis on only one sample raises major concerns whether the detected cell populations and differentially expressed genes are indeed representative and would stand up to further rigorous analyses of the additional samples analyzed. There are efficient algorithms that allow batch corrections between different samples thereby providing the basis for analysis of single cell samples isolated and analyzed from multiple individuals. It is simply not sufficient to base the entire study essentially on a single sample.

The decision to proceed using SoptSC analyses is not entirely clear. It does identify an additional population of cells that are classified by the authors as basal, however, fails to distinguish melanocytes and Langerhans and erythrocytes as separate populations, while Seurat successfully

clusters them separately, and so is arguably giving a better performance for this specific task. Regardless of the choice between clustering with Seurat or SoptSC Figure 1 is difficult to interpret. For example, the plots provided illustrating the expression profiles of markers associated with differentiation and progenitor status do not align between figure 1 D and H. It seems that there are many more cells expressing INV and K10 at high levels, as well as a more pronounced population of cells that overlap for K14 (a basal cell marker) and K10 (a differentiation marker) in plots presented in Figure 1H than in plots presented in Figure 1D. The same raw data on differential expression and levels of gene expression is presumably used to generate both plots, so it is unclear why such a discrepancy should be present. Naturally this could reflect the fact that the authors chose different color schemes to present the data. Figure 1 D displays expression data ranging from light yellow to red, while in figure H data is presented from light yellow through red to dark brown (with no scale bar for both). Therefore, as the data is presented at the moment it is difficult to interpret it in a meaningful way.

The authors then performed differential expression analyses on the individual populations that are assigned the basal identity in SoptSC and demonstrate via IHC in Figure 2 E-H that markers are expressed in subset of cells within the foreskin epidermis. This analysis needs to be supported larger images showing a bigger area of the epidermis and also quantification of the analysis. Instead of zooming in on regions like the authors have done in E-H, it would be much more informative to display the staining similar to that for K14/K10 as in Figure 2D. Importantly, from Figure 2B it is evident that some of the markers identified are not exclusive to just one population of basal cells. Co-staining for the markers in figure 2 (E-H) is a requirement alongside quantification of the analysis.

The results for the RNA analysis needs further support. Firstly, the proposed signaling analysis (Figure 3) does not provide evidence for autocrine or paracrine signaling within the epithelium. This provides the basis for hypothesizes that can then subsequently be tested functionally. Secondly, the data presented for pseudotime trajectories and velocity is not very strong and the provided supporting data based on knock down is currently not analyzed in a manner whereby the authors can relate this to the proposed trajectories. Importantly, the analyses include no quantification of phenotypes, expression analysis for markers identified in the in vivo sample, as well as data that support that the same population dynamics observed in the one in vivo samples is recapitulated in the in vitro skin reconstitution model system included here (Figure 4).

In Figure 5 the authors use PTTG1 and RRM2 KD to assess their role in skin reconstitution assays and conclude that PTTG1 as a marker of BAS1 population is essential for epidermal homeostasis unlike the other BAS populations identified. Much more sophisticated analyses are required for such statements. Again, further characterization is a requirement and importantly this should be supported by analysis of replicate samples (currently n=1).

Minor comments

Supplementary figure 1: Labels for dyes are missing

All figures - the figures are not display consistently e.g. order of cell populations, labels are missing, information needs to be obtained from previous figures.

Reviewer #1 (Remarks to the Author):

This manuscript is well written, and the data is novel with multiple detailed computational analysis performed in a statistically rigorous manner for Single Cell RNA seq data. The manuscript is of very high quality and is ready for publication.

We thank the reviewer for their detailed analysis and comments.

Reviewer #2 (Remarks to the Author):

The paper describes droplet-based scRNA-seq of primary cells from five donors. However, many of the presented analyses are based on a single donor. Supplementary Figure 4 indicates some variability in the cell clustering from different donors. For example, the GRN cluster is not detected in all of the donors. In my opinion, it would be important to show that the key findings from analyses of Library 3 are reproducible across donors. These include the expression of key marker genes (highlighted in Figure 2A) and the results of the pseudotime analysis.

We agree with the reviewer that an integrative dataset should be shown. We have included all five libraries in an integrative dataset in Supplementary Figure 4 and keratinocyte only integrative dataset combined with pseudotime in Supplementary Figure 14, which show remarkable similarity with Library 3 that we reference on page 5 paragraph 1 and page 9 paragraph 2.

We have also updated our calling of clusters in each library in Supplementary Figure 5 to depict when multiple epidermal subpopulations are combined into one cluster. We have also reduced the number of clusters in some instances based upon large distances in their respective eigengap values and/or when there are no obvious differences in their differentially expressed gene heatmaps. The clustering now shows where GRN cells cluster in each library, which can be detected in all libraries and the integrative datasets and are referenced in Supplementary Figure 5.

- On page 9 the authors state that “SoptSC unbiasedly reconstructed a putative BAS-SPN-GRN keratinocyte differentiation trajectory (Figure 4B)”. However, the figure does not clearly show GRN being the end point of the trajectory. Instead, based on 4A-B it seems that some of the GRN cells in fact precede SPN cells in pseudotime. In my opinion, the figure could be improved by addition of a panel visualizing pseudotime vs. cell identities (using the colour scheme from panel A).

We thank the reviewer for their suggestions and we now have included a Cell ID versus pseudotime plot, which shows each cluster along pseudotime with GRN at the end of the trajectory (Figure 4C).

We have also revised the pseudotime trajectory to drop the lowest probability edges per cell and presented only the high probability interactions between clusters, which now shows the highest probabilities between clusters as weighted edges and the number of cells in each cluster as dot size (Figure 4B). This analysis shows BAS-III transitioning to both SPN-I and BAS-IV and BAS-IV transitioning to SPN-II based upon their highest interaction probabilities, which are now more in line with the RNA velocity data in Figure 4E and mentioned on page 10 paragraph 1. The GRN cluster is at the end of this new pseudotime trajectory.

In addition, we show an orthogonal pseudotime method (Diffusion pseudotime; Supplementary Figure 13) that depicts the BAS-SPN-GRN trajectory with the GRN cluster at the end of this particular pseudotime and is mentioned on page 9 paragraph 2.

Finally, we have added a pseudotime trajectory for the integrative clustering of keratinocytes, which maintains the BAS-SPN-GRN trajectory but shows more complexity in the highest interaction probabilities between BAS-III, BAS-IV, SPN-I, and SPN-II (Supplementary Figure 14), also mentioned on page 9 paragraph 2.

Minor

- The cell-cell network inference analysis is rather difficult to interpret from Figure 3. From the heatmaps it would seem that the Basal cell cohort expresses on average the highest levels of genes of each pathway, yet based on the circos plots they are not among the active clusters. Could the authors please explain this discrepancy? Would it be make sense to show the expression of ligands and receptors separately?

We thank the reviewer for pointing out our previous cell-cell network wasn't as clear as we intended. The heatmaps show average gene expression for each gene in their respective clusters and was not a measure of cell-cell signaling probability. We have moved these heatmaps to Supplementary Figure 8 and clarified what the heatmaps are showing.

Finally, in the methods section it is suggested that the circos plots have directionality indicated by arrows. However, the figure is much too small to see such detail.

To improve the resolution of the circos plots and improve clarity of our cell-cell signaling interaction modeling, we completely retooled how we estimate cell-cell signaling by adding in upregulated and downregulated downstream targets to each ligand-receptor pair identified from NetPath (Kandasamy et al., 2010).

We also clustered the ligand-receptor pairs by similarity in their interaction probability scores, with the clustering shown in Figure 3A and ligand-receptor-target interaction probability scores shown in Figure 3B.

Finally, we show circos plots that represent the averaged cell-cell interaction scores at the cluster level (Figure 4C-D). This allows for larger arrows for ease in interpretation. We feel that the new panels are easier to read and provides more specific information as to which putative ligand-receptor pairs may be signaling between clusters. Because of space constraints, we show the WNT signaling pathway in Figure 3, with JAK-STAT in Supplementary Figure 9, NOTCH in Supplementary Figure 10, and TGFbeta in Supplementary Figure 11. The text has been modified on pages 7-9 to reflect these changes.

- Combining entropy and RNA velocity is an elegant and innovative approach for delineating differentiation trajectories. Unfortunately, panel 4E is too small for seeing the direction of the vectors. Please improve this as it is central information. (The same applies also to Fig. 5E)

We have now enlarged the arrows for Figure 4E and Figure 5E, and provided enlarged graphs in Supplementary Figure 15.

- P. 18 methods: “. For downstream analyses, we kept cells which met the following filtering criteria per biological replicate per condition: <6000 UMI/cell, and <10% mitochondrial gene expression. “ Is this UMI filter correct? From Supplementary Figure 2 it is evident that a significant fraction of cells has more than 6000 UMIs.

Thank you for pointing out the typo. It now reads >200 and <5000 genes/cell. We have also clarified Supplementary Figure 2 to specify the violin plots of the metrics are before quality control cutoffs.

- Figure 1C and 1G legends are identical – in the interest of clarity it would be good to point out the different clustering methods used for defining the cohorts

We have now amended the Figure legend and added whether they were generated by SoptSC or Seurat.

- Figure 1H does not seem to be cited in the text

We have now cited Figure 1H in the text.

- In the results section supervised clustering in Seurat is mentioned. But the methods section describes unsupervised clustering. Which one was used in fact?

We thank the reviewer for pointing out this area of confusion. SoptSC uses unsupervised clustering. Seurat is supervised in the sense that we placed the resolution at 0.6, which dictates the number of clusters Seurat displays. We have referenced this in the text on page 4 paragraph 2 and page 5 paragraph 2.

- Typos: P.20 “circus plots” → circos plots

We have now fixed the typos.

Reviewer #3 (Remarks to the Author):

First and foremost, the authors clearly state that they produced 5 libraries for single cell analyses, however, they have chosen to focus on one library from a single individual. Human samples are heterogenous by nature and focusing the analysis on only one sample raises major concerns whether the detected cell populations and differentially expressed genes are indeed representative and would stand up to further rigorous analyses of the additional samples analyzed. There are efficient algorithms that allow batch corrections between different samples thereby providing the basis for analysis of single cell samples isolated and analyzed from multiple individuals. It is simply not sufficient to base the entire study essentially on a single sample.

We thank the reviewer for their comments and have now included all five libraries in an integrative dataset in Supplementary Figure 4 and a keratinocyte only integrative dataset in Supplementary Figure 14, which shows remarkable similarity with Library 3. The integrative dataset has similar clusters with similar cell proportions to Library 3, with one SPN cluster splitting into two when melanocytes are removed (panel A in each figure). We do not detect significant batch effects when combining the libraries (panel B in each figure). We reference the integrative datasets and their similarity to Library 3 on page 5 paragraph 1 and page 9 paragraph two.

The decision to proceed using SoptSC analyses is not entirely clear. It does identify an additional population of cells that are classified by the authors as basal, however, fails to distinguish melanocytes and Langerhans and erythrocytes as separate populations, while Seurat successfully clusters them separately, and so is arguably giving a better performance for this specific task.

We thank the reviewer for pointing out this area of confusion. We used SoptSC because the methods employed for inference of clusters is unsupervised and SoptSC allows simultaneously inference of pseudotime, cell lineage, and cell-cell signaling within the same mathematical framework. We have clarified this on page 4 paragraph 2. Although SoptSC has already been benchmarked against a number of clustering (including Seurat) and pseudotemporal programs (Wang et al., 2019), we compared SoptSC to Seurat given the widespread use of Seurat and found they both do a pretty good job of clustering. SoptSC does a better job at clustering distinct BAS populations (which appear upon subclustering KRT14+ clusters in Seurat) and Seurat generates more SPN/GRN clusters that do not appear to be substantially distinct from each other.

Seurat does call very low populations of cells as separate clusters, however, this is dependent on the resolution that is given by the user (which we set as 0.6). Library 3, which is the one used in the main figures, is the only library with erythrocytes as a separate population, with Langerhans cells clustered in three out of the five libraries (Figure 1 and Supplementary Figure 5), which we now clarify in the text on page 5 paragraph 2. We can artificially increase the number of clusters in SoptSC and eventually observe separate clusters of Langerhans and erythrocytes in Library 3, but more clusters are generated from BAS and SPN clusters first that do not have obviously distinct gene expression profiles (data not shown). In a way, this is similar to Seurat which shows expanded SPN clusters that do not have obviously distinct gene expression profiles from each other.

Regardless of the choice between clustering with Seurat or SoptSC Figure 1 is difficult to interpret. For example, the plots provided illustrating the expression profiles of markers associated with differentiation and progenitor status do not align between figure 1 D and H.

We apologize for not making it clear enough in the original submission. Our original goal was to highlight the main populations that were clustered with SoptSC and Seurat. This meant showing the Langerhans and erythrocyte clusters called in Seurat that were not called in SoptSC. We would be happy to remove the Langerhans and erythrocyte feature plots in Figure 1H if the reviewer thinks this is a point of confusion.

It seems that there are many more cells expressing INV and K10 at high levels, as well as a more pronounced population of cells that overlap for K14 (a basal cell marker) and K10 (a differentiation marker) in plots presented in Figure 1H than in plots presented in Figure 1D. The same raw data on differential expression and levels of gene expression is presumably used to generate both plots, so it is unclear why such a discrepancy should be present. Naturally this could reflect the fact that the authors chose different color schemes to present the data. Figure 1 D displays expression data ranging from light yellow to red, while in figure H data is presented from light yellow through red to dark brown (with no scale bar for both). Therefore, as the data is presented at the moment it is difficult to interpret it in a meaningful way.

We thank the reviewer for pointing out our inconsistency with using a similar color range. We have now applied a similar color range for all feature plots. The new color scheme now shows better consistency in the expression of the selected marker genes between the two clustering programs (Figure 1 D, H).

The authors then performed differential expression analyses on the individual populations that are assigned the basal identity in SoptSC and demonstrate via IHC in Figure 2 E-H that markers are expressed in subset of cells within the foreskin epidermis. This analysis needs to be supported larger images showing a bigger area of the epidermis and also quantification of the analysis. Instead of zooming in on regions like the authors have done in E-H, it would be much more

informative to display the staining similar to that for K14/K10 as in Figure 2D. Importantly, from Figure 2B it is evident that some of the markers identified are not exclusive to just one population of basal cells. Co-staining for the markers in figure 2 (E-H) is a requirement alongside quantification of the analysis.

We thank the reviewer for suggesting ways we can better depict our data. In this resubmission, we have now quantified the immunofluorescent staining intensity of ASS1 and KRT19 at the top and bottom of the rete ridges (Figure 1K), which demonstrates significant differences in staining intensity based upon position. The quantifications are referenced on page 6 paragraph 1.

We have also quantified the distance of PTTG1 and RRM2 cells from the basement membrane, with basal and suprabasal cells serving as controls (Figure 2L), which shows distinct positioning of PTTG1 and RRM2 within the basal/suprabasal layers. The quantifications are referenced on page 6 paragraph 1.

We have also added larger representative images of ASS1, KRT19, PTTG1, and RRM2 in Supplementary Figure 6 and referenced on page 6 paragraph 1. We show co-staining of PTTG1 (BAS-I marker) and RRM2 (BAS-II marker; Figure 5K and Supplementary Figure 6) as these are the two most similar clusters in their positioning. We do observe some overlap between RRM2 and PTTG1, with distinct RRM2 only cells that have significantly distinct positioning from the basement membrane compared to PTTG1 cells (Figure 2L).

The results for the RNA analysis need further support. Firstly, the proposed signaling analysis (Figure 3) does not provide evidence for autocrine or paracrine signaling within the epithelium. This provides the basis for hypothesizes that can then subsequently be tested functionally.

We fully agree with this point. In this resubmission, we have tempered our language to indicate our cell-cell signaling network inference suggest communication within and between clusters of epidermal cell communities on page 9 paragraph 1 and throughout pages 7-9.

Secondly, the data presented for pseudotime trajectories and velocity is not very strong

We thank the reviewer for pointing out a need for more pseudotime analysis and better data presentation. We now include a Cell ID versus pseudotime plot, which shows each cluster along pseudotime (Figure 4C).

We have also revised the pseudotime trajectory to drop the lowest probability edges per cell and show only the high probability interactions between clusters, which now shows the highest probabilities between clusters as weighted edges and the number of cells in each cluster as dot size (Figure 4B). This analysis shows BAS-III transitioning to both SPN-I and BAS-IV and BAS-IV transitioning to SPN-II based upon their highest interaction probabilities, which are now more in line with the RNA velocity data in Figure 4E and mentioned on page 10 paragraph 1.

In addition, we show an orthogonal pseudotime method (Diffusion pseudotime; Supplementary Figure 13) that depicts the BAS-SPN-GRN trajectory and is mentioned on page 9 paragraph 2.

We have added a pseudotime trajectory for the integrative clustering of keratinocytes, which maintains the BAS-SPN-GRN trajectory but shows more complexity in the highest interaction probabilities between BAS-III, BAS-IV, SPN-I, and SPN-II (Supplementary Figure 14), also mentioned on page 9 paragraph 2.

Finally, we have enlarged the RNA velocity arrows for Figure 4E and Figure 5E, and provided enlarged graphs in Supplementary Figure 15.

and the provided supporting data based on knock down is currently not analyzed in a manner whereby the authors can relate this to the proposed trajectories. Importantly, the analyses include no quantification of phenotypes, expression analysis for markers identified in the in vivo sample,

We thank the reviewer for pointing out a need for more quantification of our knockdown analyses. We have now performed additional analysis of our data to include quantification of total cells per squared area (Figures 4J, 5M) and KI67+ cells (Figures 4K, 5N). This data now indicates *PTTG1* and *RRM2* knockdown both show significant differences in total number of cells per squared area and KI67+ cells per 200µm, with *PTTG1* KD the more severe of the two. This new data has been added to the text on page 13 paragraph 2.

We have also performed qRT-PCR for transcripts representing each BAS subpopulation (Figures 4L, 5O). *HELLS* KD does not show a significant change in any of the tested mRNA levels, whereas *UHRF1* KD shows significant increases in *PTTG1* and *ASS1* mRNA levels. On the other hand, *PTTG1* and *RRM2* KD both show significant changes in all the tested mRNA levels. *PTTG1* KD increased expression of *RRM2*, *GJB2*, and *PCNA* mRNA, with decreased *ASS1* mRNA. *RRM2* KD showed reductions in all tested mRNA levels. This new data has been added in the text on page 11 paragraph 2 and page 13 paragraph 2.

as well as data that support that the same population dynamics observed in the one in vivo samples is recapitulated in the in vitro skin reconstitution model system included here (Figure 4).

We thank the reviewer of bringing up this point. To help clarify this issue, we have performed qRT-PCR for transcripts representing specific keratinocyte subpopulations along Ca^{2+} -induced differentiation (Figure 5J). *PTTG1* (BAS-I) and *RRM2/PCNA* (BAS-II) mRNA expression goes down upon Ca^{2+} -induced differentiation of primary human keratinocytes, whereas *ASS1* (BAS-III) goes up during differentiation.

We also show mRNA levels of genes representing each BAS subpopulation in the *HELLS/UHRF1* KD (Figure 4L) and *PTTG1/RRM2* KD (Figure 5O). These data demonstrate that these transcripts are present in primary human keratinocytes that make up the organotypic culture and have been added in the text on page 11 paragraph 2 and page 13 paragraph 2.

In addition, we have performed scRNA-seq analysis of the human skin equivalent organotypic culture and compared them to Library 3 as shown below as a Reviewer-Only Figure. We observe each subpopulation of cells in our organotypic culture system, with BAS-I/II cells clustered together (similar to how Seurat clusters them in most libraries and SoptSC in Library 4; Supplementary Figure 5) and BAS-III and BAS-IV clustering separately. We also see additional clusters which we classify as surface ectoderm, suprabasal non-stratified, and wound re-epithelialization, potentially recapitulating the transition from a single layer culture to a stratified tissue. This data is part of a larger, rigorous analysis of different organotypic culture conditions that is currently being written up for submission elsewhere.

In Figure 5 the authors use *PTTG1* and *RRM2* KD to assess their role in skin reconstitution assays and conclude that *PTTG1* as a marker of BAS1 population is essential for epidermal homeostasis unlike the other BAS populations identified. Much more sophisticated analyses are required for such statements. Again, further characterization is a requirement and importantly this should be supported by analysis of replicate samples (currently n=1).

We thank the reviewer for pointing this out. We have tempered our language to indicate that specific genes within either BAS-I or BAS-II clusters are required for epidermal homeostasis in human organotypic cultures, and not the cluster itself, on page 13 paragraph 2.

In addition, we have increased our replicate samples for *PTTG1* and *RRM2* KD from 3 to 4, and further characterized the KDs to include quantification of total cells per squared area (Figures 5M), KI67+ cells (Figures 5N), and performed qRT-PCR for transcripts representing each BAS subpopulation (Figures 5O).

Minor comments

Supplementary figure 1: Labels for dyes are missing

We have added the labels in the legend.

All figures - the figures are not display consistently e.g. order of cell populations, labels are missing, information needs to be obtained from previous figures.

We have now clarified and consistently represented legends, ordering, and labeling in all figures.

For example, in the main figures, Figure 1C is now ordered as in Figure 1B. Figure 2B is now consistent with Figure 1C. Figure 2C is colored coded as in Figure 2B. Figure 4A, C shows labels for color coding. Figure 4H shows labels for cell lineage. Figure 5A, I show labels for color coding. Figure 5D shows labels for cell lineage inference. Figures 5F-H replaces colored dots with labels.

Reviewers' Comments:

Reviewer #2:

Remarks to the Author:

I am pleased to state that in this revised manuscript the concerns that I previously raised have been quite thoroughly addressed by the authors. Most importantly, the authors have demonstrated that the key observations are supported by each of the replicate datasets. In addition, presentation of the cell interaction analysis is much improved. I have no further reservations about the validity of the presented analyses, and in my opinion the revised manuscript is suitable for publication.

Reviewer #3:

Remarks to the Author:

In this revised version of the manuscript the data is presented more clearly, however the main concerns raised in the previous revision still stand.

Namely the entire study is still based on a sample from one individual. Though the authors show data on integrated datasets they fall short of performing the entire analyses i.e. differentially expressed genes in the different cell population clusters, predictions of cellular hierarchies, pathway activation etc. It is unclear why the authors chose not to proceed with the integrated data set for all subsequent analyses as they have all the necessary data at hand. As it stands currently the analyses of the basal cell populations is still only performed on one sample, which is simply insufficient and falls below the level of quality expected of a single cell analysis study.

The authors further go on to present signalling analyses again based exclusively on data from one library to explore possible autocrine and paracrine interactions between the cell populations, importantly, as currently presented these are merely hypotheses and the authors have not performed the necessary functional studies to test their predictions (effective inhibitors for Notch, Wnt and TGF β are readily available). It should be explicitly stated that these are hypotheses based on predictions, and not validated observations.

The data related to mouse tail epidermis distracts from the main message of the paper. Human skin does not have the pattern of ortho (K14/LRC) and parakeratotic (inv/non-LRC) epidermis observed in the tail epidermis and the comparisons consequently make no sense given that human skin is orthokeratotic.

Additional data has been provided for the KD studies performed indicating that PTTG1 and RRM2 play a role in epidermal homeostasis. The phenotypes shown are complex and show effects on cell behaviour but these studies currently do not provide any evidence for delineating the relationship between BAS1-3 populations, as argued by the authors in the discussion.

Reviewer #2 (Remarks to the Author):

I am pleased to state that in this revised manuscript the concerns that I previously raised have been quite thoroughly addressed by the authors. Most importantly, the authors have demonstrated that the key observations are supported by each of the replicate datasets. In addition, presentation of the cell interaction analysis is much improved. I have no further reservations about the validity of the presented analyses, and in my opinion the revised manuscript is suitable for publication.

We thank the reviewer for their detailed analysis and comments.

Reviewer #3 (Remarks to the Author):

In this revised version of the manuscript the data is presented more clearly, however the main concerns raised in the previous revision still stand. Namely the entire study is still based on a sample from one individual. Though the authors show data on integrated datasets they fall short of performing the entire analyses i.e. differentially expressed genes in the different cell population clusters, predictions of cellular hierarchies, pathway activation etc. It is unclear why the authors chose not to proceed with the integrated data set for all subsequent analyses as they have all the necessary data at hand.

We thank the reviewer for their comments and agree that an integrative dataset should be shown and attempted to replicate the main findings during the last revision. We now have moved those results to the main figures and performed subsequent integrative analyses and substituted those results throughout the rest of the figures and text, maintaining our main original conclusions. We have replaced Figure 1, Figure 2A-D, Figure 4A-H, Figure 5 A-I, Supplementary Figure 12 (now 11), Supplementary Figure 13 (now 12), Supplementary Figure 15 (now 13), Supplementary Figure 16 (now 14), and Supplementary Figure 19 (now 17) with the new integrative analyses and made the corresponding changes throughout the text.

The authors further go on to present signaling analyses again based exclusively on data from one library to explore possible autocrine and paracrine interactions between the cell populations, importantly, as currently presented these are merely hypotheses and the authors have not performed the necessary functional studies to test their predictions (effective inhibitors for Notch, Wnt and TGF β are readily available). It should be explicitly stated that these are hypotheses based on predictions, and not validated observations.

We thank the reviewer for their comments and have now included a statement on page 9 paragraph 1 that reads: "It should be noted that our cell-cell network inference generates hypotheses based upon predictive modeling and does not experimentally validate these events."

For the cell-cell network inference analysis, the most important parameter is gene expression. If the relevant pathway-specific gene is not expressed, the analysis cannot be done for that ligand-receptor pair. When the libraries are combined into an integrative dataset, the normalization that is done for batch correction reduces gene number and reads per cell which substantially influences the cell-cell network inference. For example, the ligand-receptor pair signaling probabilities for the JAK-STAT pathway is shown in Reviewer Only Figure 1A. The top signaling probability chart is for Library 3 and the bottom chart is the integrative dataset. The integrative dataset has lost over 90% of the ligand-receptor pairing probabilities, substantially gutting the analysis. In addition, the loss of reads per cell significantly changes the signaling probabilities as can be observed for the NOTCH pathway in Reviewer Only Figure 1B. Cluster 1 in the top and

bottom charts are quite different in their signaling probabilities, and this goes for other similar ligand-receptor paired clusters, such as Cluster 3 (top)-Cluster 4 (bottom), Cluster 4 (top)-Cluster 2 (bottom), and Cluster 5 (top)-Cluster 3 (bottom). Cluster 2 (top) is lost in the integrated dataset. For these reasons, we have maintained the original cell-cell network inference analysis for Library 3 in the main figures and added our reasoning on page 7 paragraph 1.

Reviewer Only Figure 1. Visualization of signaling probability scores of Ligand-Receptor pairs and their downstream signaling components for the **A)** JAK-STAT pathway and **B)** NOTCH pathway. Top chart features Library 3 scores. Bottom chart features the integrative dataset scores.

The data related to mouse tail epidermis distracts from the main message of the paper. Human skin does not have the pattern of ortho (K14/LRC) and parakeratotic (inv/non-LRC) epidermis observed in the tail epidermis and the comparisons consequently make no sense given that human skin is orthokeratotic.

We thank the reviewer for bringing up the differences within mouse tail epidermis, which we failed to explain in our original text. We chose the datasets by Mascré et al. 2012 and Sada et al. 2016 because they generated two of the four models of IFE differentiation and they produced corresponding datasets for their respective stem cell populations that we could compare with our BAS subpopulations. While there are clear differences between interscale (orthokeratotic – most similar to dorsal back epidermis) and scale (parakeratotic – lacks a granular layer and retains nuclei in cornified layers) IFE, clonal analysis has shown some clones originating in either region

can cross regional boundaries (Gomez et al., 2013), suggesting that they arise from similar basal populations that are likely to be differentially regulated. In addition, Sada et al. compared mouse tail epidermis to dorsal back epidermis and found similar segregation of label retaining and non-label retaining stem cells which suggest unappreciated structural similarities between both epidermal regions. We have included these references on page 12 paragraph 2. In addition, to prevent distraction from the main message of the paper, we moved these results into Supplementary Figure 17.

Additional data has been provided for the KD studies performed indicating that *PTTG1* and *RRM2* play a role in epidermal homeostasis. The phenotypes shown are complex and show effects on cell behaviour but these studies currently do not provide any evidence for delineating the relationship between BAS1-3 populations, as argued by the authors in the discussion.

We agree with the reviewer that our KD phenotypes are complex and that they only show that loss of *UHRF1*, *HELLS*, *PTTG1* or *RRM2* expression result in varying levels of epidermal phenotypes. We have attempted to clarify our language throughout the text and in the discussion to indicate that our results *suggest* that specific genes within the BAS populations are important for epidermal homeostasis in human skin equivalents and do not represent the BAS populations on a whole.

Reviewers' Comments:

Reviewer #3:

Remarks to the Author:

The manuscript have improved by integrating data from the replicates analysed in the study. I do however still have one remaining concern:

In the current version the authors on page 7 states: " We used Library 3 to generate cell-cell interaction scores because of the greater median gene number per cell (3,104 median genes per cell), an essential parameter that allows more ligand-receptor pairs to be quantified and a parameter that is reduced when all libraries are integrated because of normalization from batch correction."

It remains to be proven that the higher median gene number per cell and lack of integrated normalization and batch correction provides better molecular insight into the ligand-receptor basis for cell cell interactions. The authors could have tested this by simply generating a higher read counts for other libraries, and thereby enhancing the resolution. An alternative explanation that is not explored here is that there are significant variation between individuals and that the analysis provided here represent a snapshot of what is happening in this particular sample from this individual, but does not represent general features for how interfollicular epidermal keratinocytes are regulated.

It therefore remains unclear to this reviewer why the authors again chose to place this much emphasis on 1 sample from 1 individual (n=1- described in over 2 full pages of text 1 regular figure and 3 supplementary figures) and make general conclusions without additional follow-up experiments.

How can the authors know that the data from this one sample can be trusted and not the results from the other 4 samples. Are these speculations really necessary?

Reviewer #3 (Remarks to the Author):

The manuscript have improved by integrating data from the replicates analysed in the study. I do however still have one remaining concern:

In the current version the authors on page 7 states:" We used Library 3 to generate cell-cell interaction scores because of the greater median gene number per cell (3,104 median genes per cell), an essential parameter that allows more ligand-receptor pairs to be quantified and a parameter that is reduced when all libraries are integrated because of normalization from batch correction."

It remains to be proven that the higher median gene number per cell and lack of integrated normalization and batch correction provides better molecular insight into the ligand-receptor basis for cell cell interactions. The authors could have tested this by simply generating a higher read counts for other libraries, and thereby enhancing the resolution. An alternative explanation that is not explored here is that there are significant variation between individuals and that the analysis provided here represent a snapshot of what is happening in this particular sample from this individual, but does not represent general features for how interfollicular epidermal keratinocytes are regulated.

It therefore remains unclear to this reviewer why the authors again chose to place this much emphasis on 1 sample from 1 individual (n=1- described in over 2 full pages of text 1 regular figure and 3 supplementary figures) and make general conclusions without additional follow-up experiments.

How can the authors know that the data from this one sample can be trusted and not the results from the other 4 samples. Are these speculations really necessary?

We thank the reviewer for their comments throughout the review process and appreciate their opinion that our manuscript has improved with revision. To address whether Library 3 can generally represent all our datasets with respect to cell-cell signaling, we have now developed a consistency score that compares the strength of the ligand-receptor interactions and their directionality between clusters. Using library 3 as a reference, we compared the signaling probabilities from the other four libraries and generated consistency score graphs now presented in Supplementary Figure 7. These scores show high consistency between library 3 and the other libraries where ligand-receptor overlap exists, suggesting that the signaling interactions from library 3 generally represent the other four libraries.

Using this new analysis, we have observed that expression of the constituent ligand-receptor pair and their downstream targets are important to generate high consistency scores. This expression comes from both greater median gene number per cell and higher cell counts, which are able to sample more genes overall given the greater number of individual cell references. In light of this, we have amended our text to state: "We used Library 3 to generate cell-cell interaction scores because of the high cell count and the greater median gene number per cell (3,104 median genes per cell), allowing more ligand-receptor pairs to be quantified than when all libraries are integrated because of normalization from batch correction and showing high interaction score consistency between ligand-receptor pairs among all libraries".